# Improved cloud detection for the Aura Microwave Limb Sounder: Training an artificial neural network on colocated MLS and Aqua-MODIS data

Frank Werner[1], Nathaniel J. Livesey[1], Michael J. Schwartz[1], William G. Read[1], Michelle L. Santee[1], and Galina Wind[2,3]

[1]Jet Propulsion Laboratory, California Institute of Technology, 4800 Oak Grove Drive, Pasadena, CA 91109, USA
[2]NASA Goddard Space Flight Center, Greenbelt, Maryland, 20771, USA
[3]SSAI Inc., Lanham, Maryland, 20706, USA

**Correspondence:** Frank Werner (frank.werner@jpl.nasa.gov)

**Abstract.** An improved cloud detection algorithm for the Aura Microwave Limb Sounder (MLS) is presented. This new algorithm is based on a feedforward artificial neural network and uses as input, for each MLS limb scan, a vector consisting of 1,710 brightness temperatures provided by MLS observations from 15 different tangent altitudes and up to 13 spectral channels in each of 10 different MLS bands. The model has been trained on global cloud properties reported by Aqua's Moderate Resolution Imaging Spectroradiometer (MODIS). In total, the colocated MLS-MODIS data set consists of $162,117$ combined scenes sampled on 208 days over 2005–2020. A comparison to the current MLS cloudiness flag used in "Level 2" processing reveals a huge improvement in classification performance. For previously unseen data, the algorithm successfully detects $> 93\%$ of profiles affected by clouds, up from $\approx 16\%$ for the Level 2 flagging. At the same time, false positives reported for actually clear profiles are comparable to the Level 2 results. The classification performance is not dependent on geolocation, but slightly decreases over low-cloud cover regions. The new cloudiness flag is applied to determine average global cloud cover maps over 2015–2019, successfully reproducing the spatial patterns of mid-level to high clouds seen in MODIS data. It is also applied to four example cloud fields to illustrate its reliable performance for different cloud structures with varying degrees of complexity. Training a similar model on MODIS-retrieved cloud top pressure ($p_{CT}$) yields reliable predictions with correlation coefficients $> 0.82$. It is shown that the model can correctly identify $> 85\%$ of profiles with $p_{CT} < 400\,\mathrm{hPa}$. Similar to the cloud classification model, global maps and example cloud fields are provided, which reveal good agreement with MODIS results. The combination of cloudiness flag and predicted cloud top pressure provides the means to identify MLS profiles in the presence of high-reaching convection.

# 1 Introduction

The impact of clouds on Earth's hydrological, chemical, and radiative budget is well established (e.g., Warren et al., 1988; Ramanathan et al., 1989; Stephens, 2005). With the introduction of satellite imagery, the first studies of cloud observations from space concentrated on the determination of cloud cover (e.g., Arking, 1964; Clapp, 1964). After the advent of multispectral satellite radiometry, retrievals of increasingly comprehensive suites of cloud macrophysical, microphysical, and optical characteristics were developed (e.g., Rossow et al., 1983; Arking and Childs, 1985; Minnis et al., 1992; Kaufman and Nakajima,

1993; Han et al., 1994; Platnick and Twomey, 1994). Such efforts require a reliable cloud detection prior to the actual retrieval process. Conversely, there are remote sensing applications where clouds, rather than being the subject of interest, are a source of artifacts that negatively impact the observation of desired geophysical variables. For land and water classifications, clouds and cloud shadows represent unusable data points that need to be detected accurately and discarded (e.g., Ratté-Fortin et al., 2018; Wang et al., 2019). Because of the similar spectral behavior of aerosols and clouds, and their complicated interactions,

deriving reliable aerosol properties from space requires careful cloud detection with high spatial resolution (e.g., Varnai and Marshak, 2018). Instruments operating in the ultraviolet to infrared spectral wavelength ranges cannot penetrate any but the optically thinnest clouds. As a result, retrievals of atmospheric composition in the presence of clouds are severely limited.

Approaches to cloud detection from satellite-based imagers are characterized by varying levels of complexity, from simple thresholding and contrast methods to multi-level decision trees (e.g., Ackerman et al., 1998; Ackerman et al., 2008; Zhao and

Di Girolamo, 2007; Saponaro et al., 2013; Werner et al., 2016). In recent years fast machine learning algorithms have been employed to detect cloudiness based on observed spatial and spectral patterns (e.g., Saponaro et al., 2013; Jeppesen et al., 2019; Sun et al., 2020). Regardless of the technique, each algorithm must be designed purposefully and with the respective application in mind, as discussed in Yang and Di Girolamo (2008).

The Aura Microwave Limb Sounder (MLS), which has provided global retrievals of atmospheric constituent profiles from

$\sim 10$ km to $\sim 90$ km since 2004, operates at frequencies from 118 GHz to 2.5 THz. In this spectral range clouds are much more transparent than at shorter wavelengths, and the impact on the measured radiances is low. Only clouds with high liquid and/or ice water content reaching altitudes of $\sim 9$ km and higher can significantly impact the sampled radiances. The current MLS "Level 2" cloud detection algorithm is based on the computation of cloud induced radiances ($T_{\mathrm{cir}}$), which represent the difference between individual observations and calculated clear sky radiances (Wu et al., 2006). The latter are derived after

the retrieval of the other MLS data products. To first order, scattering from thick clouds diverts a mix of large upwelling radiances, from lower in the atmosphere, and smaller downwelling radiances, from above, into the MLS raypath. Accordingly, for sufficiently thick clouds within the MLS field of view, $T_{\mathrm{cir}}$ will be positive for limb pointings above an altitude of $\sim 9$ km, where non-scattered limb views are characterized by low radiances. Conversely, $T_{\mathrm{cir}}$ will be negative below $\sim 9$ km, where non-scattered signals would otherwise be large. In the MLS Level 2 processing, if the absolute value of $T_{\mathrm{cir}}$ exceeds predefined

detection thresholds, then the respective profile is flagged as being influenced by high or low clouds. The thresholds are set for individual retrieval phases and spectral bands; e.g., for MLS bands 7–9, around a center frequency of 240 GHz, radiances are flagged where $T_{\mathrm{cir}} > 30$ K or $T_{\mathrm{cir}} < -20$ K. Subsequently, separate retrieval algorithms deduce ice water content and path from

the $T_{\mathrm{cir}}$ information (Wu et al., 2008). Note that in earlier phases of the MLS Level 2 processing, a similar scheme, computing clear sky radiances based on preliminary retrievals of temperature and composition, is used to identify MLS radiances that have been significantly affected by clouds and discard them in the final atmospheric composition retrievals.

The focus for the Level 2 flagging is on identifying cases where clouds impact the MLS signals sufficiently to potentially affect the MLS composition retrievals. However, the reliance on global, conservatively defined thresholds will inherently induce uncertainties in the current cloud detection scheme. For optically thinner clouds, where $T_{\mathrm{cir}}$ values are close to but do not exceed the prescribed thresholds, the current cloud flag will provide a false clear classification. Improvements to the current cloud detection scheme could allow: (i) a comprehensive uncertainty analysis of the retrieval bias induced by clouds, (ii) more reliable MLS retrievals in the presence of clouds, where a potential future correction of MLS radiances could account for the cloud influence, (iii) identification of composition profiles that can be confidently considered to be completely clear sky, and (iv) the reliable identification of profiles in the presence of high-reaching convection. Points (iii) and (iv) have the potential to enable new science studies. For example, a reliable cloud mask for individual MLS profiles would enable more comprehensive analysis of lower-stratospheric water vapor enhancements associated with overshooting convection. Currently, studies of these events rely on computationally expensive colocation of water vapor profiles with cloud properties from different observational sources (e.g., Tinney and Homeyer, 2020; Werner et al., 2020; Yu et al., 2020).

This study describes the training and validation of an improved MLS cloud detection scheme employing a feedforward artificial neural network ("ANN" hereinafter). This algorithm is derived from colocated MLS samples and MODIS cloud products and is designed to classify clear and cloudy conditions for individual MLS profiles. Two specific goals are set for the new algorithm: (i) detection of both high (e.g., cirrus and cumulonimbus) and mid-level (e.g., stratocumulus and altostratus) clouds, and (ii) detection of less opaque clouds containing lower amounts of liquid or ice water. Observed cloud variables, used to train the ANN, are provided by the Moderate Resolution Imaging Spectroradiometer (MODIS) aboard NASA's Aqua platform. Of the major satellite instruments, Aqua MODIS observations are most suitable for this study, as they provide operational cloud products on a global scale that are essentially coincident and concurrent with the MLS observations.

The manuscript is structured as follows: section 2 describes both the MLS and MODIS data used in this study. Then a short introduction to the general setup of a feedforward ANN is given in section 3.1, followed by specifics on the output (section 3.2), input (section 3.3), and the training and validation procedure (section 3.4) of the developed models. Results from applying the cloud detection algorithm to MLS data are given in section 4, which includes a statistical comparison of the prediction performance between the Level 2 and ANN results (section 4.1), a discussion about ANN performance for uncertain cases (section 4.2), a global performance evaluation and cloud cover analysis (section 4.3), and four example scenes contrasting the performance of the Level 2 flag and the new algorithm for different cloud fields in section 4.4. The performance of the subsequent cloud top pressure predictions is presented in section 5, which comprises an evaluation of the prediction performance and an assessment of the model's ability to detect high clouds (section 5.1), global maps (section 5.2), and four example scenes comparing the ANN predictions to the MODIS results (section 5.3). The main conclusions and a brief summary are given in section 6.

## 2 Data

Aura MLS samples brightness temperatures ($T_B$) in five spectral frequency ranges around 118, 190, 240, 640, and 2,500 GHz (Waters et al., 2006) (the latter, measured with separate, independent optics, was deactivated in 2010 and is not considered here). Multiple bands, consisting of 4–25 spectral channels, cover each of these frequency ranges; see Table 4 in Waters et al. (2006) and Figure 2.1.1 in Livesey et al. (2020). The exact position of the specific bands was chosen based on the different absorption characteristics of the various atmospheric constituents that MLS observes. MLS makes $\approx 3500$ daily vertical limb scans (called major frames; MAFs), each consisting of 125 minor frames (MIFs) that can be associated with tangent pressures ($p_{tan}$) at different altitudes in the atmosphere. These observations provide the input for retrievals of profiles of a wide-ranging set of atmospheric trace gas concentrations. The respective Level 2 Geophysical Product (L2GP) files also report a status diagnostic for every MLS profile, which includes flags indicating high and low cloud influence. The most recent MLS dataset is version 5; however, at the time the ANN was being developed, reprocessing of the entire MLS record to date with the v5 software had not yet been completed. Accordingly, L2GP cloudiness flags in this study are provided by the version 4.2x data products (Livesey et al., 2020), and v4.2x is also the source for the Level 1 radiance measurements used herein. Note that the sampled radiances are identical between the two versions, while revisions to the atmospheric composition retrieval algorithms yield subtle differences in the derived cloudiness flags. The spatial resolution of MLS Level 2 products varies from species to species, but typical values are 3 km in the vertical and $5 \times 500$ km in the cross-track and along-track dimensions. The distance along the orbit track between adjacent profiles is $\approx 165$ km.

Global cloud variables used in this study are provided by retrievals from the Aqua-MODIS instrument, which precedes the Aura overpass by about 15 minutes. However, because of the differences in their viewing geometries, the true time separation between MLS and MODIS measurements is substantially smaller than 15 minutes (see section 3.2). MODIS collects radiance data from 36 spectral bands in the wavelength range 0.415–14.235 $\mu$m. For a majority of the channel observations and subsequently retrieved cloud properties, the spatial resolution at nadir is $1,000$ m, although the pixel dimensions increase towards the edges of a MODIS granule. Each granule has a viewing swath width of $2,330$ km, enabling MODIS to provide global coverage every two days. More information on MODIS and its cloud product algorithms (the current version is Data Collection 6.1) is given in Ardanuy et al. (1992); Barnes et al. (1998); Platnick et al. (2017). Each pixel, $j$, within a MODIS granule reports a value for the cloud flag, a cloud top pressure ($p_{CT}^j$), cloud optical thickness ($\tau^j$), and effective droplet radius ($r_{eff}^j$). These last two variables are used to derive the total water path ($Q_T^j$), which contains both the liquid and ice water path and characterizes the amount of water in a remotely sensed cloud column. It can be calculated following the discussions in Brenguier et al. (2000); Miller et al. (2016):

$$Q_T^j = \Gamma \cdot \rho^j \cdot \tau^j \cdot r_{eff}^j, \tag{1}$$

where $\rho^j$ is the bulk density of water in either the liquid or ice phase (following the cloud phase retrieval for pixel $j$), and the factor $\Gamma$ accounts for the vertical cloud structure. For vertically homogeneous clouds it can be shown that $\Gamma = 2/3$.

Table A1 in the Appendix lists the 208 days that comprise the global data set used in this study. It consists of twelve random days annually, one for each month, for the years between 2005 and 2020, as well as one additional day each year that forms a set of consecutive days. This brings the yearly coverage to thirteen days.

## 3 Artificial neural network

This section provides details about the ANN setup and training. Here, we constructed and trained a multilayer perceptron, which is a subcategory of feedforward ANNs that sequentially connects neurons between different layers. In a feedforward ANN information only gets propagated forward through the different model layers and is not directed back to affect previous layers. An introduction to multilayer perceptrons is given in section 3.1. The output vector containing the labels (i.e., the binary cloud classifications) based on a colocated MLS-MODIS data set, and the features (i.e., the input matrix), which consist of MLS $T_B$ observations, are described in sections 3.2 and 3.3, respectively. The choice of hyperparameters, the training setup, and the validation results from the algorithm are provided in section 3.4.

The weights that connect the input to the output data are determined by the "Keras" library for Python (version 2.2.4; Chollet et al., 2015) with "TensorFlow" (version 1.13.1) as the backend (Abadi et al., 2016).

### 3.1 Algorithm description

Figure 1 illustrates the general setup of a simplified multilayer perceptron that contains four layers, and is purely instructional. The complete model setup is more complex and is discussed in sections 3.2–3.4. The input layer (shown in blue) consists of $m = 3$ vectors that contain selected MLS brightness temperatures $\mathbf{T}_{B1}$, $\mathbf{T}_{B2}$, and $\mathbf{T}_{B3}$. The input layer is succeeded by two hidden layers (shown in green) with two neurons each ($N_{h1-1}$ and $N_{h1-2}$, as well as $N_{h2-1}$ and $N_{h2-2}$) and the respective bias vectors ($\mathbf{B}_1$ and $\mathbf{B}_2$). The following output layer (shown in orange) consists of a single vector ($\mathbf{L}$; containing the predicted labels) and a corresponding bias ($B_L$). The brightness temperature vectors ($\mathbf{T}_{Bi}$; $i = 1, 2, 3$) used as input for the ANN are provided by $T_B$ observations in selected channels, bands, and minor frames. They are of length $n$, which describes the number of scalar MLS observations ($T_{Bi}^j$). This means, that $i = 1, 2, 3$ brightness temperatures were sampled by MLS at $j = 1, \ldots, n$ major frames. Similarly, there is a scalar label $L^j$ for each MAF, so $\mathbf{L}$ is also of length $n$. All bias vectors are initialized to 1.

At each neuron $N_{h1-k}$, $k = 1$–$2$ in the first hidden layer, a scalar value $\gamma_{1-1}^j$ and $\gamma_{1-2}^j$ for each of the $j$ MAFs is calculated:

$$\gamma_{1-1}^j = B_{1-1} \cdot \omega_{0,1} + T_{B1}^j \cdot \omega_{1,1} + T_{B2}^j \cdot \omega_{2,1} + T_{B3}^j \cdot \omega_{3,1} \tag{2}$$

$$\gamma_{1-2}^j = B_{1-2} \cdot \omega_{0,2} + T_{B1}^j \cdot \omega_{1,2} + T_{B2}^j \cdot \omega_{2,2} + T_{B3}^j \cdot \omega_{3,2}. \tag{3}$$

Here, the weights $\omega$ connect the observed brightness temperatures (and the bias) to the neurons in the first hidden layer. $\gamma_{1-1}^j$ and $\gamma_{1-2}^j$ are subsequently modified by an activation function, which introduces non–linearity into the neuron output. The hyperbolic tangent activation function is applied, which is shown to be very efficient during training because of its steep gradients (e.g., LeCun et al., 1989; LeCun et al., 1998) and yields new values $\Gamma_{1-1}^j$ and $\Gamma_{1-2}^j$. For the second hidden layer, the

scalar neuron values at $N_{h2-k}$, $k = 1$–$2$ for each MAF $j$ are derived as:

$$\gamma_{2-1}^{j} = B_{2-1} \cdot \varpi_{0,1} + \Gamma_{1-1}^{j} \cdot \varpi_{1,1} + \Gamma_{1-2}^{j} \cdot \varpi_{2,1} \tag{4}$$

$$\gamma_{2-2}^{j} = B_{2-2} \cdot \varpi_{0,2} + \Gamma_{1-1}^{j} \cdot \varpi_{1,2} + \Gamma_{1-2}^{j} \cdot \varpi_{2,2}, \tag{5}$$

where the weights $\varpi$ connect the neuron output from the first hidden layer, as well as the bias, to the neurons in the second hidden layer. As before, these scalar neuron values are transformed by the hyperbolic tangent activation function, which yields the transformed neuron values $\Gamma_{2-1}^{j}$ and $\Gamma_{2-2}^{j}$.

Finally, the neuron output from $N_{h2-1}$ and $N_{h2-2}$ is connected to the single vector $\mathbf{L}$ in the output layer via weights $\Omega$. For each MAF $j$ the respective scalar value $\lambda^{j}$ is calculated as:

$$\lambda^{j} = B_{L} \cdot \Omega_0 + \Gamma_{2-1}^{j} \cdot \Omega_1 + \Gamma_{2-2}^{j} \cdot \Omega_2. \tag{6}$$

We aim for a binary, two-class cloud classification setup (i.e., either cloudy or clear designations) and information about the probability for each predicted class. As a result, the softmax function normalizes the $\lambda^{j}$ results at the output layer. The softmax activation function is identical to the logistic sigmoid function for a binary, two–class classification setup. This means that the predicted neuron output in the output layer is calculated as:

$$\hat{L}^{j} = \frac{1}{1 + \exp\left(-\lambda^{j}\right)}. \tag{7}$$

The model for the cloud top pressure prediction uses a simple pass-through of the neuron output to the output layer. The ideal weights in Eqs. (2), (3), (4), (5) and (6) need to be derived iteratively by evaluating a loss function ($\chi$), which is the log–loss function (or cross-entropy) in the classification setup. If $L^{j}$ and $\hat{L}^{j}$ are the individual elements of the two output vectors $\mathbf{L}$ and $\hat{\mathbf{L}}$ (i.e., the prescribed and currently predicted labels), $\chi$ for two classes is defined as:

$$\chi = -\sum_{j=1}^{n} L^{j} \cdot \ln\left(\hat{L}^{j}\right) + \left(1 - L^{j}\right) \cdot \ln\left(1 - \hat{L}^{j}\right) + R. \tag{8}$$

Here, $R$ is an optional regularization term that is used to control the stability of the respective model. Note that in case of $L^{j} = 0$ or $\hat{L}^{j} = 0$ an infinitesimal quantity $\epsilon \approx 0$ is added to the respective label to avoid the undefined $\ln 0$. Conversely, the model for the cloud top pressure prediction minimizes the mean squared error. The "Keras" algorithm includes multiple optimizers to solve Eq. (8) in a numerically efficient way. The exact setup and choice of hyperparameters need to be determined carefully via cross-validation during the training process (see section 3.4).

## 3.2 The labels from colocated MLS-MODIS cloud data

Training data for the output vector $\mathbf{L}$, which contains the prescribed labels for Eq. (8), is provided by the MODIS C6.1 data set described in section 2. The reported MODIS cloud products are first colocated with individual MLS profiles.

An example MLS orbit on 19 May 2019 is shown in Figure 2a. Each blue dot represents one of the $\approx 3500$ daily profiles sampled by MLS. Note that there are three latitudinal ranges (in the tropics, as well as northern and southern mid-latitudes),

where the ascending and descending orbits cross multiple times a day. Since the inclination angle of Aura is close to $90°$, both polar regions contain more MLS profiles than other locations.

The illustration in Figure 2b depicts how colocation is performed. If $n_{per}$ is the number of MODIS pixels (gray shaded squares) within a $1° \times 1°$ box (in latitude and longitude; blue box) around an MLS profile (blue "x"), then each of the $n_{per}$ pixels reports a cloudiness flag, as well as a total water path ($Q_T^j$) and a cloud top pressure ($p_{CT}^j$), with $j = 1, 2, ..., n_{per}$ denoting the individual pixels within the $1° \times 1°$ box. Note that for legibility the cloud properties of only three MODIS pixels are shown. For the respective MLS profile, these parameters are aggregated to more general cloud statistics consisting of the cloud cover ($C$) within the $1° \times 1°$ box, as well as the median total water path ($Q_T$) and median cloud top pressure ($p_{CT}$). Note that no significant decrease in classification performance is observed for varying aggregation scales between $0.5° \times 0.5°$ and $2° \times 2°$.

Figure 2c shows the global distribution of sample frequencies for the colocated MLS-MODIS data set within grid boxes of length $15° \times 15°$ (latitude and longitude). While not every grid box contains the same number of profiles, each area contains at least 2,100 MLS-MODIS samples. A maximum in sample frequency is observed over the regions with denser MLS coverage around the poles.

The aggregated profile-level cloud statistics are used to define the observed clear sky and cloudy conditions. All profiles that are characterized by $C \geq 2/3$, $p_{CT} < 700\,\text{hPa}$, and $Q_T > 50\,\text{g m}^{-2}$ are labeled as cloudy, while profiles with $C < 1/3$ and $Q_T < 25\,\text{g m}^{-2}$ are considered to be associated with clear sky samples. While the cloud cover threshold is somewhat arbitrary, the $p_{CT}$ limit for cloudy observations and the $Q_T$ thresholds are carefully selected. The large opacity of the atmosphere for longer path lengths means that MLS shows almost no sensitivity towards clouds with $p_{CT} \geq 700\,\text{hPa}$ (see section 3.3). This upper pressure limit, which in the 1976 US Standard Atmosphere (COESA, 1976) is located at an altitude of $\sim 3\,\text{km}$, is around the lower limit of observed cloud tops of mid-level cloud types (e.g., altostratus, altocumulus). The $10^{th}$ and $25^{th}$ percentiles of all profiles containing clouds within the $1°$ perimeter, regardless of $C$, are $Q_T \approx 25\,\text{g m}^{-2}$ and $Q_T \approx 50\,\text{g m}^{-2}$, respectively. These definitions have an additional benefit: they almost evenly split the data set into cloudy and clear sky profiles ($52.0\%$ and $48.0\%$, respectively), which improves the reliability of the trained weights for the cloud classification.

Naturally, these definitions leave some profiles undefined (e.g., those with $C$ in the range 1/3–2/3). These profiles (about the number of the combined cloudy and clear classes) cannot be included in the training of the ANN, as they lack a prescribed label. The discussion in section 4.1 provides an analysis of the ANN performance for a redefined classification based on a simple threshold of $C = 0.5$ (in addition to a positive $Q_T$) to distinguish between cloudy and clear sky profiles.

Figure 2d shows the global distribution of sample frequencies for the training data set, which comprises the clear sky and cloudy labels defined earlier. Here, the observed patterns depend strongly on the MODIS-observed cloud conditions (see section 4.3 for more information). Regions with comparatively low cloud cover (most of the African continent, as well as Australia and Antarctica) and those with increased occurrences of high and mid-level clouds (mostly over land) show higher sample frequencies compared to areas over the oceans. Three regions with low sample frequencies, west of South America, Africa, and Australia, stand out. Those areas are characterized by increased $C$ of low clouds of up to $80\%$ (e.g., Muhlbauer et al., 2014). Similar patterns are observed over the North Pacific and Atlantic Oceans, albeit to a lesser extent. Those MLS

profiles are influenced by clouds that are either too low or exhibit $C < 1/3$, and are therefore not included in the training data set (i.e., are part of the undefined class mentioned earlier).

It is important to note that the difference in viewing geometry between MLS and MODIS (i.e., limb geometry versus nadir viewing) induces a considerable degree of uncertainty in the colocation. While it is reasonable to assume that the majority of a potential cloud signal (or lack thereof) will come from the $1° \times 1°$ box around the respective MLS profile, there are certain scenarios that will lead to a false classification. The most likely such scenario consists of an MLS line-of-sight that passes through a high-altitude cloud before a clear sky $1° \times 1°$ box. Here, MLS will detect a strong cloud signal, even though the nadir-viewing MODIS instrument does not record any cloudiness at the location of the respective MLS profile. Less likely is the scenario of a very low-altitude cloud located right after (in terms of an MLS line-of-sight) a clear sky $1° \times 1°$ box. This would also result in a false cloud classification (if the MODIS observations are taken as reference). However, because of the increase in atmospheric opacity, the sensitivity of the MLS instrument towards signals further along the line-of-sight decreases, and it is less likely that MLS would detect these cloud signals in any case. One contributor to the overall uncertainty that is of less concern is the time difference between the Aqua and Aura orbits ($\approx 15$ minutes). Because MLS looks forward in the limb, the temporal discrepancy between the sampling of individual MLS profiles and the colocated MODIS pixels is in the range of 0.6–1.4 minutes. The results presented in section 3.4 illustrate that by training the ANN with a large data set, as well as cross-validating the training results against a large number of random validation data, the contributions of uncertainties associated with colocation (both in space and time) can be considered small and do not overly impact the reliability of the cloud detection algorithm.

The reader is also reminded of the fact that the proposed ANN schemes will try to reproduce, as best as they can, the MODIS-retrieved cloud variables. Those parameters, however, have their own uncertainties and biases, and the ANN will inherently learn those MODIS-specific characteristics. As a result, the ANN predictions should not be considered the true atmospheric state. Instead, they represent a close approximation of the observed values in the colocated MLS-MODIS data set.

### 3.3 The input matrix from MLS brightness temperature observations

Figures 3a-c show the spectral behavior of $T_B$ sampled in MLS bands 2, 33, and 14 at MIF=15, which on average corresponds to $p_{tan} \sim 576\,\text{hPa}$ (at an altitude of $\sim 4.5\,\text{km}$ in the 1976 US Standard Atmosphere). In this section we mostly omit the superscript "j" to indicate the statistical analysis of all $T_B^j$ in the respective band ($j = 1, 2, \cdots, n$). The median $T_B$ for profiles associated with clear sky (orange) and cloudy conditions (blue), based on the classifications from the colocated MLS-MODIS data set described in section 3.2, are shown by the solid lines and circles. The shaded orange and blue areas indicate the interquartile range (IQR; $75^{\text{th}}$-$25^{\text{th}}$ percentile of data points) of clear and cloudy profiles. Data are from profiles sampled in the latitudinal range of $-30°$ to $+30°$.

Median clear sky profiles exhibit consistently larger $T_B$ than cloudy observations, with differences of up to 10 K. This behavior confirms the findings in Wu et al. (2006), where ice clouds at an altitude of 4.7 km reduce band 33 $T_B$ at the lower minor frames (i.e., larger $p_{tan}$). The IQR ranges of the two different data sets are very close for band 2 observations (i.e., within 1–2 K), while there is overlap for the $T_B$ sampled in bands 33 and 14.

To illustrate the reduced sensitivity of MLS to signals from very low clouds, the median $T_{\mathrm{B}}$ from profiles with $p_{\mathrm{CT}} \geq 700\,\mathrm{hPa}$ is shown in green (for clarity the corresponding IQR is omitted). These profiles behave similarly to clear sky observations, and the difference in median $T_{\mathrm{B}}$ is less than 1 K.

Figures 3d-f illustrate the spectral behavior of $T_{\mathrm{B}}$ sampled at MIF=33, which corresponds to an average $p_{\mathrm{tan}}$ of $\sim 200\,\mathrm{hPa}$ (at an altitude of $\sim 12\,\mathrm{km}$ in the 1976 US Standard Atmosphere). Similar to the results for the lower MIF, a clear separation between median $T_{\mathrm{B}}$ from clear sky and cloudy ($100\,\mathrm{hPa} \leq p_{\mathrm{CT}} < 700\,\mathrm{hPa}$) profiles is observed, while those profiles associated with low clouds ($p_{\mathrm{CT}} \geq 700\,\mathrm{hPa}$) again behave similarly to clear samples. For observations from bands 2 and 33, the cloudy profiles show significantly higher $T_{\mathrm{B}}$. Again, this confirms the reported behavior in Wu et al. (2006), who found an increase in band 33 $T_{\mathrm{B}}$ for cloudy conditions compared to the clear background. Conversely, band 14 observations behave similarly to those sampled at MIF=15, and the cloudy profiles exhibit lower $T_{\mathrm{B}}$.

The significant contrast in median $T_{\mathrm{B}}$ between clear sky and cloudy profiles, especially for band 2 and partly for band 33, might suggest the possibility of a simple cloud detection approach via thresholds. However, the respective IQR ranges often overlap, which indicates that a simple $T_{\mathrm{B}}$ threshold would miss about $25\%$ of both the clear and the cloudy data. Moreover, the behavior illustrated in Figure 3 is specific to the latitudinal range of $-30°$ to $+30°$. For higher latitudes, changes in atmospheric temperature and composition yield a noticeable decrease in the observed contrast, while close to the poles the clear sky profiles almost always have lower $T_{\mathrm{B}}$ than the cloudy observations (even at the lower MIFs). A more sophisticated classification approach, with $T_{\mathrm{B}}$ samples from additional MLS bands and minor frames, is necessary to derive a more reliable global cloud detection.

Table 2 details the MLS bands, as well as their associated channels and MIFs, that comprise the $m \times n$ input matrix for the ANN. The input matrix consists of $m$ different $T_{\mathrm{B}}^{\mathrm{j}}$, sampled in individual channels (within the respective MLS bands) and MIFs, at $n$ different times. To reduce the computational costs during the training of the model, not all MLS observations are considered. Instead, ten different bands are chosen in total. Those are bands 2, 3, 6; bands 7, 8, 33; and bands 10, 14, 28 for the 190, 240, and 640\,GHz spectral regions, respectively. These bands were carefully selected after a statistical analysis of the altitude-dependent contrast in observed $T_{\mathrm{B}}$ between clear and cloudy profiles. This contrast is generally low (in the range of 1 K) for the observations from the 118\,GHz region, so only band 1 from this receiver is included in the model input. For most of the ten bands, every second channel is included in the input (except for band 33, which only has 4 channels in total), while considering every third MIF in the range 7–49 yields decent vertical resolution between 15\,hPa (for the highest altitudes) and 150\,hPa (at the lowest altitudes). Overall, the input matrix for the training and validation of the ANN is of shape $1,710 \times 162,117$; i.e., it consists of $m = 1,710$ different features ($T_{\mathrm{B}}^{\mathrm{j}}$ at different frequencies and altitudes) from $n = 162,117$ MAFs (either classified as clear sky or cloudy).

## 3.4 Training and validation

The "Keras" python library provides convenient ways to manage the setup, training, and validation of ANN models. The optimal weights for Eqs. (2), (3), (4), (5) and (6) are derived in four steps: (i) defining an independent test data set, which comprises $10\%$ of the clear and cloudy cases, and will be used to evaluate the final model, (ii) determining the most appropriate

hyperparameters via $k$-fold cross-validation, (iii) training and validating a number of different models with the best set of hyperparameters on multiple, random splits between training and validation data sets, and (iv) comparing the performance scores for the different model runs to evaluate the stability of the approach and pick the best set of weights.

    The hyperparameters to be determined are (i) the number of hidden layers, (ii) the number of neurons per hidden layer, (iii) the optimizer for the cloud classification, (iv) the mini-batch size, (v) the learning rate, and (vi) the value for the weight

decay (i.e., the L2 regularization parameter). The number of hidden layers and neurons impact the complexity of the model. The choice of optimizer controls how fast and accurately the minimum of the loss function in Eq. (8) is determined, based on different feature sets and minimization techniques. During each iteration the model computes an error gradient and updates the model weights accordingly. Instead of determining the error gradient from the full training data set, our models only use a random subset of the training data (called a mini-batch) during each iteration. This not only speeds up the training process,

but also introduces noise in the estimates of the error gradient, which improves generalization of the models. The learning rate controls how quickly the weights are updated along the error gradient. Thus, the size of the learning rate affects the speed of convergence (higher is better) and ability to detect local minima in the loss function (lower is better). Meanwhile, L2 regularization is one method to specify the regularization term $R$ in Eq. (8), where the sum of the squared weights is multiplied with the L2 parameter:

$$R = \text{L2} \cdot \sum \omega^2 + \varpi^2 + \Omega^2. \tag{9}$$

Note that for clarity we omitted the indices for the weights in Eq (9). The amount of regularization is directly proportional to the value of the L2 weight decay parameter. Regularization usually improves generalization of the models. More information about ANN hyperparameters and their impact on the reliability of model predictions can be found in, e.g., Reed and Marks (1999) and Goodfellow et al. (2016).

The optimal number of hidden layers and neurons was determined to be in the range 1–2 and 100–1,200 (in increments of 100), respectively. The mini-batch size alternated between $2^5$ and $2^{13}$. The learning rate was varied between $10^{-6}$ and $10^{-2}$ in increments of 2 levels per decade; the L2 parameter covered a range between $10^{-7}$ and $10^{-1}$ (as well as L2 $= 0$).

    The number of epochs (i.e., the number of iterations during the training process) is not considered an important hyperparameter for this study. Instead, the models are run with a large number of epochs, and the lowest validation loss is recorded, so

an increase in validation loss during the training (i.e., cases where the model is overfitting the training data at some point) has no impact on the overall performance evaluation. Note that the lowest validation loss usually occurred after $\sim 2{,}000 - 3{,}000$ epochs for both the cloud classification and $p_{\text{CT}}$ prediction. No obvious increase in validation loss was observed, even for a large number of epochs.

### 3.4.1   Determining the hyperparameters

At first, the remaining $90\%$ of data points (after removing the random test data set) are randomly shuffled and split into $k = 4$ parts. Subsequently, one of the four parts is used as the validation data set, and the other three are used to train the ANN with a certain set of hyperparameters. Here, each of the $1{,}710$ features is individually standardized, i.e., each input variable is

transformed to have a mean value of 0 and unit variance. This step is essential for a successful ANN training, as the individual features are characterized by different dynamic ranges. Meanwhile, the labels for clear and cloudy profiles are simply set to

0 and 1, respectively. For the $p_{CT}$ models the labels are simply set to the respective $p_{CT}$ values. After model convergence and determination of a set of performance scores, the model is discarded and a different set of three parts is used for training (the remaining fourth part is again used for validation). After cycling through each of the four parts (and recording four sets of performance scores), the set of hyperparameters is changed and the process begins anew. An evaluation of each set of performance scores, for each set of hyperparameters, reveals the appropriate setup for the ANN.

The performance scores employed for the cloud classification training are three commonly used binary classification metrics, based on the calculation of a confusion matrix $\mathbf{M}$ for the two classes (i.e, clear sky and cloudy profiles). If $tp$ and $tn$ are the number of true positives and negatives, respectively, and $fp$ and $fn$ are the number of false positives and negatives, respectively, then the confusion matrix is defined as:

$$\mathbf{M} = \begin{pmatrix} tp & fp \\ fn & tn \end{pmatrix}. \tag{10}$$

From $\mathbf{M}$ the accuracy ($Ac$), F1 score ($F1$) and Matthews correlation coefficient ($Mcc$) can be derived as:

$$Ac = \frac{tp + tn}{tp + tn + fp + fn} \tag{11}$$

$$F1 = \frac{2 \cdot tp}{2 \cdot tp + fp + fn} \tag{12}$$

$$Mcc = \frac{tp \cdot tn - fp \cdot fn}{\sqrt{(tp + fp) \cdot (tp + fn) \cdot (tn + fp) \cdot (tn + fn)}}. \tag{13}$$

While $Ac$ quantifies the proportion of correctly classified samples, $F1$ describes the harmonic mean value between precision
(proportion of true positives in the positively predicted ensemble, i.e., the ratio of $tp$ to $tp$+$fp$) and recall (proportion of correctly predicted true positives, i.e., the ratio of $tp$ to $tp$+$fn$). Generally, $F1$ assigns more relevance to false predictions and is more suitable for imbalanced classes, where the respective data sizes vary significantly. All elements of the confusion matrix are important in determining the $Mcc$, which yields values between $-1$ and $1$ and thus is analogous to a correlation coefficient.

The performance evaluation for the $p_{CT}$ prediction application, on the other hand, is based on the Pearson product-moment
correlation coefficient ($r$) and root-mean-square deviation (RMSD).

For the cloud classification application, this analysis revealed that models using one hidden layer slightly outperformed those with two hidden layers. The number of neurons per hidden layer had a negligible impact, as long as the number was larger than 200. However, the models with 800 and 900 neurons exhibited average $Ac$ values that were 0.0002 higher than those of other setups. We ultimately set it to $856$, which corresponds to the average between the number of nodes in the input
and output layers (i.e., $1,710$ and 1, respectively). The Adam optimizer with a learning rate of $10^{-5}$ yielded the overall best validation scores for the cloud classification. Note that we applied the Adam optimizer with the standard settings described in the "Keras" documentation. The best L2 parameter and mini-batch size values were found to be $50^{-4}$ and 1,024 (i.e., $0.8\%$ of the training data), respectively. Note that while the choice of L2 had the largest influence on model performance, the impact of the mini-batch size was comparable to the number of neurons (as long as it was $> 2^6$).

For the $p_{CT}$ prediction, two-layer models noticeably outperformed single layer ones, as the drop in average $r$ was $> 0.01$. Again, the number of neurons had only a minimal impact on model performance, with variations in $r$ of $\approx 0.02$. However, models with 800–1,000 neurons performed best, so we again set this number to 856. The best optimizer, learning rate, L2 parameter, and mini-batch size were found to be Adam, $10^{-4}$, $50^{-4}$, and 1,024, respectively.

### 3.4.2   Validation statistics

Due to randomness during the assignment of individual observations to either the training or validation data set, developing a single model might result in evaluation scores that are overly optimistic or pessimistic. By chance, the most obvious cloud cases (e.g., $C = 1$ and very large $Q_T$ values) might have ended up in the validation data set, or vice versa, and the trained weights might be inappropriate. Moreover, a large disparity in validation scores for multiple models might be indicative of an ill-posed problem, where the MLS observations do not provide a reasonable answer to the cloud classification problem. Therefore,
developing multiple models with a reasonable split of training and validation data, as well as careful monitoring of the spread in validation scores, is imperative. In this study, 100 different models are developed. Before each model run, the data set (minus the test data set) is randomly shuffled and split into training and validation data. The splits between training/validation/test data are set to $70/20/10\%$. The hyperparameters are identical for each model. As mentioned earlier, each model is run with a large number of epochs, and the weights associated with the lowest validation loss are recorded. Training of these 100 models took
$\sim$1 day.

    The output of each cloud classification model is a cloudiness probability ($P$) between 0 (clear) and 1 (cloudy). Note that throughout this study we simply group each prediction in either the clear or cloudy class, i.e., MAFs with predicted probabilities $0 \leq P < 0.5$ are considered to be sampled under clear sky conditions, while MAFs with $0.5 \leq P \leq 1$ are considered to be cloudy. The one exception is the discussion in section 4.2, where the actually predicted $P$ are employed to study the ANN
performance for undefined cloud conditions (with respect to the clear sky and cloudy definitions presented in section 3.2).

    A summary of the derived prediction statistics is shown in Figure 4a. Each histogram shows the average percentage of correctly predicted clear sky (i.e., $tn$, orange shading) and cloudy (i.e., $tp$, blue shading) labels for all 100 validation data sets. Also shown are the percentages of false classifications (the blue and orange lines for $fn$ and $fp$, respectively). The gray shaded horizontal areas at the top of each histogram illustrate the standard deviation for each class, calculated from the 100 validation
data sets. The average percentage of correct clear sky and cloudy predictions is $93.7\%$ and $93.2\%$, respectively, while a false cloudy or clear sky prediction occurs for $6.3\%$ and $6.8\%$ of profiles in the validation data. The standard deviation for all four groups is $0.2\%$.

    Figure 4b shows a scatter plot of all $Mcc$ values as a function of $F1$. Even though the $Mcc$ penalizes false classifications more severely than $F1$, a high $r = 0.97$ is observed. Moreover, there is little variability in the 100 derived binary statistical
metrics, with average $Ac$, $F1$, and $Mcc$ values of $0.934 \pm 0.001$, $0.937 \pm 0.001$, and $0.868 \pm 0.003$. These results illustrate that the derived models are well suited to predict cloudiness for new MLS data (i.e., measurements not involved in the training of the models) and that the trained weights are very stable (i.e., all models exhibit very similar binary statistics, regardless of the respective training or validation data set).

Similarly, histograms of $r$ and RMSD (referring to the relationship between the predicted and MODIS results for the validation data), as well as the regression between the two variables, are shown in Figures 4c-d, respectively. Average values of $0.819 \pm 0.001$ ($r$) and $80.268 \pm 0.160$ hPa (RMSD) are observed. The correlation between the two parameters is $r = -0.77$.

Given the statistical robustness of the results, the model with the highest $Mcc$ and lowest RMSD provide the ANN weights for cloud classification and $p_{CT}$ prediction in this study, respectively.

## 4    Cloud detection: Results and examples

This section includes a detailed comparison between the predicted cloud classifications from the current MLS v4.2x and the new ANN-based algorithms in section 4.1, followed in section 4.2 by a discussion of predicted cloudiness probabilities that illustrates the performance of the new ANN cloud flag for less confident cases (i.e., those outside of the training, validation, and test data sets). This section also presents an analysis of the latitudinal dependence of the ANN performance and derived global cloud cover statistics in section 4.3, as well as a close-up look at cloudiness predictions for some example scenes over both the North American and Asian monsoon regions (section 4.4).

Note that a comparison between v4.2x and ANN results will naturally favor the ANN predictions, particularly any comparison made with reference to MODIS observations. Evaluating the performance of each cloud flag is based on the respective agreement to the MODIS-observed cloud conditions. However, the ANN is designed to replicate the MODIS results, while the v4.2x algorithm is not aware of the MODIS data set (including its uncertainties and biases).

### 4.1    Prediction performance of current L2GP and new ANN cloud flag

The analysis in section 3.4 indicates that the ANN setup can reliably reproduce the cloudiness conditions identified by the colocated MLS-MODIS data set. Figure 5 provides a closer look at the performance of the new ANN-based and v4.2x cloud flags for all $n = 32, 425$ $(16, 211)$ profiles associated with either the clear sky or cloudy class in the validation (test) data set.

Figures 5a and b present the percentage of correctly classified (blue) and falsely classified (orange) cloudy validation profiles, as determined by the cloudiness definition for the colocated MLS-MODIS data set described in section 3.2. The frequency of predicted labels from the (a) new ANN-based algorithm and (b) v4.2x cloud flag are shown as a function of $Q_T$. Note that because of the general cloudiness definition, only those profiles with $Q_T > 50$ g m$^{-2}$ are considered (see section 3.2). The flags predicted by the ANN correctly classify $93.3\%$ of the cloudy profiles. In particular, the thickest clouds, those with $Q_T \geq 1,000$ g m$^{-2}$, are detected in $78.0\%$ of cases. Conversely, the current v4.2x status flag only detects $15.6\%$ of the cloudy profiles. A peak of $15.4\%$ of clouds is missed for low $Q_T$, where the ANN performs significantly better. This is understandable, as the current v4.2x status flags for high and low cloud influences should only be set for profiles where the extinction along the line-of-sight is large enough to be attributed to a fairly thick cloud. However, even for very large $Q_T \geq 1,000$ g m$^{-2}$, only $25.8\%$ of the cloudy profiles are detected.

Histograms for clear sky observations in the validation data set as a function of $C$ are presented in Figures 5c and d. Only $5.7\%$ of clear profiles are falsely classified as cloudy by the new ANN algorithm, while the current v4.2x status flag mislabels

6.2% of these profiles. Most of the clear observations occur for very low values of $C < 0.05$, of which the ANN and v4.2x flags detect 50.4% and 48.5%, respectively. Note that the slightly larger fraction of false positives from the v4.2x flag is not necessarily incorrect, i.e., there might actually be clouds in the line-of-sight of one or more MLS scans associated with the respective profiles. They might, however, be well before (very high clouds) or past (very low clouds) the tangent point and outside of the $1° \times 1°$ box defined in section 3.2.

Similar histograms for the test data set are shown in Figures 5e-h. The ANN correctly identifies 95.0% and 96.2% of the cloudy and clear cases, respectively, as well as 76.6% of the $Q_\mathrm{T} \geq 1{,}000\,\mathrm{g\,m}^{-2}$ and 51.0% of the $C < 0.05$ profiles. The respective fractions detected by the current v4.2x status flag are 15.4%, 93.8%, 26.6%, and 48.2%.

Table 3 gives an overview of the confusion matrix elements for each cloud flagging scheme, as well as metrics to evaluate binary statistics. For the validation data the new ANN algorithm yields values of $Ac = 0.94$, $F1 = 0.94$, and $Mcc = 0.87$ ($Ac = 0.96$, $F1 = 0.96$, and $Mcc = 0.91$ for the test data), confirming the reliable classification performance shown in Figure 5. The v4.2x flag yields low binary performance scores of $Ac = 0.53$, $F1 = 0.26$, and $Mcc = 0.15$ for the validation data ($Ac = 0.55$, $F1 = 0.25$, and $Mcc = 0.15$ for the test data), mainly due to the low fraction of true positives.

## 4.2 Probabilities for different cloud conditions

The clear sky and cloudy classes defined in section 3.2 leave a number of profiles unaccounted for (i.e., neither clear sky nor cloudy), such as those with $1/3 \leq C < 2/3$ or $p_\mathrm{CT} \geq 700\,\mathrm{hPa}$. While it is reasonable to only train the model on the confidently clear and cloudy conditions, it is essential to understand the ANN performance for the undefined, in-between cases.

Figure 6a shows average ANN-predicted cloudiness probabilities as a function of $C$ and $Q_\mathrm{T}$ with no restrictions on $p_\mathrm{CT}$. Data that the ANN was trained on are excluded from this analysis. Figure 6b illustrates the distribution when $P$ values are distributed into four groups: confidently clear ("Conf. Clr."; $P < 0.25$), probably clear ("Prob. Clr."; $0.25 \leq P < 0.5$), probably cloudy ("Prob. Cld."; $0.5 \leq P < 0.75$), and confidently cloudy ("Conf. Cld."; $P \geq 0.75$). The previously defined clear sky and cloudy regions are indicated by the white and black dashed lines, respectively. Profiles with low $C < 1/3$ and $Q_\mathrm{T} < 25\,\mathrm{g\,m}^{-2}$, regardless of $p_\mathrm{CT}$, are characterized by the lowest $P$ values, reliably reproducing the clear sky class defined in section 3.2. Meanwhile, almost all profiles with $C > 0.7$ are flagged to be probably cloudy ($P > 0.5$). However, only profiles that also have $Q_\mathrm{T} > 100\,\mathrm{g\,m}^{-2}$ are reliably predicted to have $P > 0.75$. The less-confident identification of the $Q_\mathrm{T} > 100\,\mathrm{g\,m}^{-2}$ cases reflects the fact that many of them have low cloud tops, $p_\mathrm{CT} \geq 700\,\mathrm{hPa}$, and are thus not readily observed by MLS. As noted in section 3.3, these profiles exhibit similar spectral behavior to clear ones, and the ANN is expected to miss most of these clouds. With increasing $Q_\mathrm{T}$, even profiles with smaller cloud fractions (as little as $C = 0.25$) are flagged as cloudy. Note that the $P$ results become noisy for very large $Q_\mathrm{T} > 500\,\mathrm{g\,m}^{-2}$, conditions that are only observed for less than 4% of the total samples ($< 1\%$ for $Q_\mathrm{T} > 1000\,\mathrm{g\,m}^{-2}$).

In order to evaluate the ANN performance when more of these uncertain cases are encompassed, we included in Table 3 a comparison of the binary performance scores for a redefined set of the cases classified as clear and cloudy according to less conservative thresholds for the cloud cover and the total water path ($C < 0.5$ and $Q_\mathrm{T} < 25\,\mathrm{g\,m}^{-2}$ for clear sky profiles, $C \geq 0.5$ and $Q_\mathrm{T} \geq 25\,\mathrm{g\,m}^{-2}$ for cloudy profiles). No limitations on $p_\mathrm{CT}$ are imposed. These changes increase the number of profiles

from $n = 48,636$ (validation and test data) to $n = 214,805$ profiles. Again, samples from the training data set are excluded. Due to the looser definitions, there is a significant drop in performance scores, which can mostly be attributed to a lower true positive rate (i.e., cloud detection) of $0.58$ and $0.05$ for the ANN classification and v4.2x, respectively. The fraction of false positives (i.e., false prediction of cloudiness for actually clear profiles) remains basically unchanged (changes of $\approx +0.04$ and $-0.01$ for the ANN and v4.2x flags, respectively). This means that even with a looser cloudiness definition, the ANN does not yield a multitude of false cloud classifications; rather, the algorithm fails to detect a larger fraction of cloudy profiles. As a consequence of the reduced true positive rates for the modified class definitions, the derived $F1$ for the ANN score is reduced to $0.58$ (from $\approx 0.94$), while $F1$ for the current v4.2x flag drops from $\approx 0.26$ to $0.09$. This is almost exclusively due to an inability to detect lower-level clouds. As demonstrated in section 3.3, MLS cannot distinguish between clear sky and cloud signals if $p_{CT} \geq 700\,\mathrm{hPa}$. Adding a threshold of $p_{CT} < 700\,\mathrm{hPa}$ to the loosened definitions, the performance for the now $n = 89,697$ profiles is much closer to the one from the validation and test data set. Here, the ANN and v4.2x classifications exhibit $Ac = 0.90$, $F1 = 0.90$, $Mcc = 0.81$ and $Ac = 0.54$, $F1 = 0.22$, and $Mcc = 0.13$, respectively.

## 4.3 Geolocation-dependent performance and global cloud cover distribution

The spectral behavior for clear sky and cloudy profiles shown in Figure 3 only applies for observations made in the latitudinal range of $-30°$ to $+30°$. As mentioned in section 3.3, the contrast between the two classes of data decreases for increasing latitude. While the analysis in section 4.1 illustrates that the new ANN-based cloud classification can reliably identify cloudy profiles (based on the definitions in section 3.2), it is important to make sure that there is no latitudinal bias in the predictions, i.e., assuring that the algorithm performance is good for MLS observations at all latitude bands.

Calculated $F1$ determined from the ANN model setup is shown in Figure 7a for different regions of the globe. Statistics are calculated in grid boxes that cover an area of $15° \times 15°$ (latitude and longitude) and include on average 168 profiles. High values $F1 > 0.85$ are observed for most regions; however, areas with generally low cloud cover (over Africa and Antarctica, as well as west of South America and Australia, see Figure 7e) exhibit slightly lower classification performance, indicated by the light blue and green colors. Here, reduced sample statistics yield a less reliable $F1$ metric, as the number of profiles per grid box is as low as 18. Further analysis shows that the reduced $F1$ scores within these grid boxes are exclusively due to an increase in false negatives, i.e., the model misses some cloudy profiles. Overall, the average observed $F1$ is $0.91 \pm 0.11$.

In contrast to the results for the ANN algorithm, there is a more noticeable latitudinal dependence for the performance of the current v4.2x algorithm, illustrated in Figure 7b. $F1$ values can be as high as $0.67$ in the tropics and $< 0.25$ everywhere else. Occasional gaps, especially over the polar regions, are due to a failed $F1$ calculation. Here, the denominator in Eq. (12) becomes 0, i.e., the v4.2x flag only reports clear sky classifications. The average observed $F1$ is $0.23 \pm 0.16$.

As the prediction performance is high for a majority of geographical regions, the ANN algorithm is applied to derive global cloud cover maps, based solely on the MLS observed $T_B$ and the calculated model weights. A map of cloudiness from all MLS profiles sampled over 2015–2019, averaged within $3° \times 5°$ (latitude and longitude) grid boxes, is shown in Figure 7c. Note that this data set includes more than 6 million MLS profiles, while only 65 days in the 5-year span were part of the training data. Profiles are considered to be cloudy when predicted $P \geq 0.5$. Three large-scale regions close to the equator show

the largest average cloud covers with $C > 80\%$ (dark orange colors): (i) an area over the northern part of South America, (ii) central Africa, and (iii) a large band encompassing the Maritime Continent. Large zonal bands of $C \approx 60\%$ are observed in the mid-latitudes of both hemispheres. Conversely, large areas of low $C < 20\%$ are observed west of the North American, South American, and African continents, as well as over Australia, northern Africa, and Antarctica. The derived cloud covers, as well as the observed spatial patterns of mid to high clouds, agree well with those reported in King et al. (2013) and Lacagnina and Selten (2014).

As before, we are interested in comparing the results of the new ANN classification to the ones from the current v4.2x cloud flag. Therefore, a similar map of derived global cloud cover from the current v4.2x cloud flag is shown in Figure 7d. In contrast to the ANN results, v4.2x suggests $C < 32\%$ almost everywhere. This behavior is consistent with the focus of the v4.2x classification, where only very opaque clouds around $\sim 300\,\mathrm{hPa}$ are flagged. The global patterns identified by the new ANN flag are reproduced, albeit with much lower results for $C$. However, the v4.2x flag yields a global maximum of $C > 72\%$ over Antarctica. Here, the new ANN flag reports $C$ as low as $3\%$. This behavior in the v4.2x cloud flag is a well-understood feature caused by misinterpretation of radiances that are reflected by the surface (W. G. Read, *personal communications*, 2021). Here, the unique combination of high topography and low optical depth makes Antarctica one of the few places where MLS can observe the Earth's surface.

Figures 7e-f show similar cloud cover maps generated from Aqua-MODIS observations. Due to the size of that data set and the high computational costs, only samples from 2019 are included here. The cloud cover maps were generated considering cloud mask flag values of 0 and 1 (confident cloudy and probably cloudy) as defined in Menzel et al. (2008). All available 1 km-resolution MODIS cloud mask data were considered. The aggregation used the high-resolution cloud top pressure product, not generally available as a global aggregation. This cloud top pressure product, however, is the one utilized by retrievals of MODIS cloud optical properties. Such custom aggregation thus ensures the maximum dataset consistency across variables. While all clouds are considered in the map in panel e, only clouds with $p_{\mathrm{CT}} < 700\,\mathrm{hPa}$ are included to derive $C$ in panel f. It is obvious that including clouds with $p_{\mathrm{CT}} \geq 700\,\mathrm{hPa}$ dramatically increases the derived cloud covers. Due to the reduced sensitivity towards such clouds (see the discussion in section 3.3), the cloud covers predicted by the ANN are much closer to the MODIS results that do not include low clouds. Nonetheless, the ANN-derived $C$ are, on average, $\sim 9\%$ higher than the MODIS results, suggesting that MLS is able to detect some of the lower clouds with $p_{\mathrm{CT}} \geq 700\,\mathrm{hPa}$. This behavior is also illustrated in the example scenes in Figures 8–9 in section 4.4. In comparison, there is much poorer agreement between the MODIS and v4.2x results, with v4.2x on average $\sim 26\%$ lower than MODIS.

This analysis indicates that the new ANN algorithm can produce considerably more reliable cloud classifications than the v4.2x MLS cloud flag, on a global scale.

## 4.4 Example scenes

The analysis in the previous sections centered on statistical metrics and the reproduction of large-scale, global cloud patterns. There, the cloud flag based on the new ANN algorithm yields reliable results, both in comparison to the current v4.2x status flag and as a standalone product. However, a more qualitative assessment of the model performance for individual cloud scenes

provides additional confidence in the technique, as well as insights into the classification performance for different cloud types. Again, profiles are flagged as cloudy when $P \geq 0.5$.

Figure 8 shows two example cloud fields over the North American monsoon region. During the summer months of July and August, this area is characterized by the regular occurrence of mesoscale convective systems that can occasionally overshoot into the lowermost stratosphere, where the sublimation of ice particles can lead to local humidity enhancements (Anderson et al., 2012; Schwartz et al., 2013; Werner et al., 2020). Observed $p_{CT}$ and $Q_T$ derived from Aqua MODIS observations over the first example scene, sampled on 31 August 2017, are shown in Figures 8a and 8b, respectively. The MLS overpass is

illustrated in gray transparent circles. A cloud system with $p_{CT} < 500\,\mathrm{hPa}$ exists in the northern part of the scene, with the lowest $p_{CT} \sim 200\,\mathrm{hPa}$. The MLS track passes some smaller cloud clusters characterized by large $Q_T$, which are indicated in yellow. In the south, low clouds with $Q_T = 50 - 450\,\mathrm{g\,m^{-2}}$ are observed. The new ANN and current v4.2x cloud flags are shown in Figures 8c-d. The ANN algorithm flags every profile in the northern part of the scene as cloudy, while also detecting the very low clouds in the south. Conversely, the classifications from the current v4.2x flag identify a cloud influence for a single

MLS profile in the north, which happens to actually be over an area with low $Q_T$. A second example cloud field is shown in Figures 8e-h. This scene consists of clouds all along the MLS track and large areas with elevated $Q_T$ up to $1{,}000\,\mathrm{g\,m^{-2}}$. Note that there is a gap in the MLS track, where the level 2 products are screened out, according to the rules in the MLS quality document (Livesey et al., 2020). The ANN algorithm correctly determines that every profile along the path was sampled under cloudy conditions. However, even for the very high clouds that contain large water abundances, the v4.2x algorithm only

occasionally flags the respective profiles as cloudy. In the northern part of the track, the flag actually alternates between clear sky and cloudy classifications.

    Similarly, Figure 9 shows two example cloud fields over the Asian summer monsoon region, which also regularly contains overshooting convection from mesoscale cloud systems. The first scene, shown in Figures 9a-d, displays a mix of different cloud conditions. There are high clouds with $p_{CT} < 350\,\mathrm{hPa}$ and $Q_T = 50 - 450\,\mathrm{g\,m^{-2}}$ in the northern part, a large clear sky

area in the middle, and then a mix of very high and low-level clouds in the south that exhibits low $Q_T$ and likely represents a multi-layer cloud structure with thin cirrus above boundary layer clouds. The new ANN-based flag successfully detects both the northern and southern cloud fields, while the current v4.2x flag only detects a single profile with cloud influence. The last example scene, illustrated in Figures 9e-h, similarly displays a mix of low, mid-level, and high clouds. As expected, the current v4.2x algorithm only flags a single profile as influenced by high clouds (in the south of the scene). However, the ANN

algorithm detects the mid-level clouds in the North, as well as the mix of cloud types in the South of the scene. In those places where MODIS mostly captured either low boundary layer clouds (yellow colors) or the cloud property retrieval failed (very low $Q_T$), the ANN associates the respective profiles with the clear sky class.

    Note that the two example scenes in Figure 9 represent previously unseen data for the ANN, i.e., the models were not trained on these MLS-MODIS observations.

## 5  Predicting cloud top pressure: Results and examples

The results in section 4 illustrate that the proposed ANN algorithm can successfully detect the subtle cloud signatures in the spectral $T_\mathrm{B}$ profiles shown in Figure 3. For many MLS bands, the differences between cloudy and clear sky $T_\mathrm{B}$ are usually in the range of just a few Kelvin, and the spectral behavior heavily depends on the respective MIF (i.e., pressure level at the tangent point of each scan). This section demonstrates how this behavior can be used in a similar ANN setup to infer the MODIS-retrieved $p_\mathrm{CT}$. Here, our goal is to reliably differentiate between mid- to low-level clouds and high-reaching convection with $p_\mathrm{CT} <\approx 350\,\mathrm{hPa}$. As mentioned in the introduction, not only can these high clouds impact the MLS retrieval of atmospheric constituents, but they can also breach the tropopause and inject ice particles into the lowermost stratosphere.

This section presents a statistical performance evaluation of the $p_\mathrm{CT}$ prediction in section 5.1, a global analysis of $p_\mathrm{CT}$-distributions in section 5.2, as well as a close-up look at $p_\mathrm{CT}$ predictions for the same example scenes over the North American and Asian monsoon regions that were shown earlier (section 5.3).

Similar to the cloud classification analysis, a comparison between v4.2x and ANN prediction performance will favor the ANN results, since the ANN is designed to replicate the MODIS observations.

### 5.1  Performance evaluation

Joint histograms of observed and predicted $p_\mathrm{CT}$ for all cloudy profiles in the validation and test data set are presented in Figures 10a and b, respectively. While there is a fair amount of scatter, the majority of data points are close to the 1:1 line. This is illustrated by the envelope indicated by the white dashed line, which is defined by the 5[th] and 95[th] percentiles of predicted $p_\mathrm{CT}$ for each observed $p_\mathrm{CT}$-bin (i.e., the envelope indicates where $90\%$ of predicted $p_\mathrm{CT}$ are). High values of $r = 0.825$ and $r = 0.839$, with RMSD values of 79.2 hPa and 76.9 hPa, are observed for the two data sets. However, a decline in ANN performance is noticeable for observed $p_\mathrm{CT} > 400\,\mathrm{hPa}$, where predictions for $p_\mathrm{CT} > 600\,\mathrm{hPa}$ exhibit an average underestimation of 126 hPa ($19.2\%$). This is consistent with the findings presented in section 3.3, which showed a reduced sensitivity of MLS observations to low-level clouds. Conversely, the average difference between predictions and observations is $+25\,\mathrm{hPa}$ ($9.5\%$) for MODIS-retrieved $p_\mathrm{CT} < 400\,\mathrm{hPa}$.

Histograms of the difference between predicted and observed $p_\mathrm{CT}$ for profiles in the validation and test data sets are shown in Figure 10c. The two distributions look almost identical and are centered around a difference of $-8$ and $-10\,\mathrm{hPa}$, respectively. For the validation data set, $65.6\%$ ($88.0\%$) of predictions are within 50 hPa (100 hPa) of the MODIS observations, while $66.9\%$ and $88.0\%$ of profiles in the test data set are within these ranges.

As mentioned in the introduction, we are mostly interested in the ability to detect high clouds with $p_\mathrm{CT} < 400\,\mathrm{hPa}$. Not only can these clouds affect the MLS radiances and retrievals, they can also impact water vapor (e.g., Werner et al., 2020; Tinney and Homeyer, 2020) and $HNO_3$ (e.g., Wurzler et al., 1995; Krämer et al., 2006) concentrations in the upper troposphere and lower stratosphere. Figure 10d shows the percent of profiles in the combined validation and test data set, where the ANN correctly reproduces the MODIS-observed cloud top pressure for thresholds of $p_\mathrm{CT} < 400, 350$, and 300 hPa. To provide a comparison to the current v4.2x algorithm performance, we simply calculated the percent of successful cloud detection for each of these $p_\mathrm{CT}$-

thresholds. The ANN correctly identifies $85.4, 80.0,$ and $78.5\%$ of the profiles with $p_{CT} < 400, 350,$ and $300\,\mathrm{hPa}$, respectively. In contrast, the v4.2x flag only detects $8.5, 8.6,$ and $8.7\%$ of these profiles.

The analysis in this section reveals that the ANN setup can predict the MODIS-retrieved $p_{CT}$ with reasonable accuracy, which provides the ability to reliably identify high clouds with $p_{CT} < 400\,\mathrm{hPa}$.

## 5.2  Geolocation-dependent performance

Similar to the cloud classification analysis presented earlier, it is important to understand the geolocation-dependent prediction performance of the $p_{CT}$ model. Figure 11a shows the global distribution of derived $r$ between the observed and predicted $p_{CT}$. Profiles from both the validation and test data sets are considered. Statistics are calculated within $15° \times 15°$ grid boxes (latitude and longitude) that contain an average of 116 cloudy profiles (following the definition in section 3.2). The average correlation coefficient in each grid box is $r = 0.75$, and strong correlations, $r > 0.80$, are recorded within all latitude ranges. However, areas with weaker correlation, $r \approx 0.4 - 0.7$, (light blue and green colors) appear to coincide with regions of low cloud cover (see Figure 7). Further analysis shows that the decreased model performance in these areas can almost exclusively be attributed to uncertainties in the prediction for clouds with $p_{CT} > 400\,\mathrm{hPa}$ (not shown). This relationship between model performance and $C$ is confirmed in Figure 11b, which illustrates the global distribution of the RMSD. Increased values are primarily observed over regions with low $C$; e.g., the highest RMSD of $181.6\,\mathrm{hPa}$ (bright yellow color) is observed west of the South American continent, which exhibits some of the lowest $C$ globally (see Figure 7). Similarly, RMSD$> 100\,\mathrm{hPa}$ are observed over Antarctica, Australia, off the coast of Africa and South America, as well as over northeastern Greenland.

Global distributions of the average MODIS-retrieved $p_{CT}$ and the predicted ANN results are shown in Figures 11c and d, respectively. The ANN can reliably recreate the patterns observed by MODIS, with high $p_{CT}$ in the high latitudes, mid-level clouds over the Southern Ocean and northern mid-latitudes, and low $p_{CT}$ over the tropics and subtropics. Especially the region with low $p_{CT} < 250\,\mathrm{hPa}$ over Southeast Asia is well reproduced by the ANN.

Again, there is particular interest in the ability of the ANN to identify high clouds with $p_{CT} < 400\,\mathrm{hPa}$. Figure 10d indicated that, overall, the $p_{CT}$ model can reliably identify profiles associated with high clouds. Figure 11e provides information about the global distribution of successful high cloud detections. The ANN correctly predicts $p_{CT} < 400\,\mathrm{hPa}$ for $> 80\%$ of profiles within grid boxes in the latitude range $-60°$ to $+60°$. Here, the average fraction of correct predictions is $85.6\%$. However, outside of that range (i.e., in the high latitudes) the average of correct classifications per grid box is only $47.7\%$. It is likely that the model simply did not learn the respective patterns associated with high clouds in these regions, where only $5.6\%$ of the global $p_{CT} < 400\,\mathrm{hPa}$ observations occur (at least according to the combined validation and test data set).

Figure 11f presents a similar map of the fraction of successful $p_{CT} < 400\,\mathrm{hPa}$-detections based on the current v4.2x algorithm. Overall, the ANN dramatically outperforms the v4.2x flag, which on average only identifies $20.8\%$ of the respective profiles within each grid box. A few areas over Antarctica are the exception, where the current algorithm manages to recognize $100\%$ of the respective profiles with $p_{CT} < 400\,\mathrm{hPa}$. This success, however, is likely a coincidence and can be attributed to the misinterpretation of radiances that are reflected by the surface. This behavior also caused the high $C$ values in the re-

gion, as shown in Figure 7d. As mentioned earlier, these samples only represent a small fraction of the total occurrence of $p_{CT} < 400\,\text{hPa}$; excluding these areas from the statistics causes the average v4.2x performance to drop by only $0.8\%$.

## 5.3 Example scenes

Similar to the analysis in section 4.4, comparisons between maps of MODIS-retrieved and predicted $p_{CT}$ for individual cloud fields provide a qualitative assessment of the model performance. MODIS results from the same four example scenes that were previously shown in Figures 8-9 are presented in Figures 12a, c, e, and f. The respective ANN predictions are shown in Figures 12b, d, f, and h. The first two scenes are sampled over the North American monsoon region, the second two over the Asian summer monsoon anticyclone.

The first example (panels a and b) consists of high clouds in the northern part of the scene, with the lowest $p_{CT} \sim 200\,\text{hPa}$ around $40°$ latitude. The ANN reliably reproduces the MODIS results and predicts the highest clouds at the right position. Mixed results are achieved for the very low clouds in the south, which are outside of the MLS-detectable pressure range. For these two profiles, the ANN predicts $p_{CT} = 585$ and $434\,\text{hPa}$. While clearly too low, the model successfully associates these samples with low clouds. The second scene (panels c and d) is characterized by high $C$ values throughout, with low to mid-level clouds in the very north and a complicated mix of different cloud types throughout the rest of the scene. Not surprisingly, the ANN identifies all but three profiles to be associated with medium to high clouds. Here, even small occurrences of high clouds in the perimeter of an MLS profile yields a low $p_{CT}$ prediction.

Three samples in the vicinity of mid-level clouds are visible in the northern part of the third scene (panels e and f), as well as two profiles above very low and two profiles above high clouds in the south. While the ANN is not able to detect the $p_{CT} > 700\,\text{hPa}$-region, it successfully predicts clouds with $p_{CT} = 343 - 511\,\text{hPa}$ northward of $35°$ latitude and $p_{CT} < 206\,\text{hPa}$ in the south. Finally, another complicated scene is depicted in panels g and h. The two southernmost profiles have a MODIS-observed $p_{CT}$ of $390\,\text{hPa}$ and $285\,\text{hPa}$, which is accurately reproduced by the ANN. Predicted $p_{CT}$ for the three northernmost profiles agree similarly well with the observations. However, the ANN predictions are too low for profiles between $25°$ and $30°$ latitude, and too high for the lone cloudy profile around $33°$ latitude.

As noted in sections 5.1-5.2, the performance for $p_{CT}$ predictions seems to decline with an increase in cloud top pressure, consistent with the reduced contrast between clear sky and cloudy $T_B$ around $p_{CT} \sim 700\,\text{hPa}$, as shown in Figure 3.

## 6 Summary and conclusions

The current MLS cloud flags, reported in the Level 2 Geophysical Product files of version 4.2x, are designed to identify profiles that are influenced by significantly opaque clouds, with the main goal being to identify cases where retrieved composition profiles may have been adversely affected either by the clouds or by the steps taken in the retrieval to exclude cloud-affected radiances. In this study, we present an improved cloud detection scheme based on a standard multilayer perceptron, a subcategory of feedforward artificial neural networks (ANNs). It applies a softmax activation function in the output layer for binary classifications (i.e., clear sky or cloudy), while a log–loss function is minimized to determine the model weights. A second

setup, which applies a linear output in the output layer and determines the model weights by minimizing the mean squared error, is used to produce a cloud top pressure ($p_{\mathrm{CT}}$) estimate from MLS radiances that approximates the MODIS retrievals.

This new algorithm is shown to not only reliably detect high and mid-level convection containing even small amounts of cloud water, but also to distinguish between high-reaching and mid- to low-level convection.

To train the ANN models we colocated global MLS brightness temperatures ($T_{\mathrm{B}}$), sampled on 208 days between 2005 and 2020, with nadir-viewing MODIS-retrieved cloud properties aggregated within a $1° \times 1°$ box (in latitude and longitude) around each MLS profile. This yielded a median cloud cover ($C$), $p_{\mathrm{CT}}$, and cloud water path ($Q_{\mathrm{T}}$) associated with each of the $162,117$

MLS scans in the colocated data set. These variables are used to discriminate clear sky ($C < 1/3$ and $Q_{\mathrm{T}} < 25\,\mathrm{g\,m^{-2}}$) from cloudy ($C \geq 2/3$, $100\,\mathrm{hPa} \leq p_{\mathrm{CT}} < 700\,\mathrm{hPa}$, and $Q_{\mathrm{T}} > 50\,\mathrm{g\,m^{-2}}$) profiles. Overall, the input variables for the ANN consist of $1,710$ MLS-observed $T_{\mathrm{B}}$ from different spectral bands, channels, and minor frames (i.e., views at different altitudes in the atmosphere). After setting aside $10\%$ of the data to serve as an independent test data set, comprehensive testing and cross-validation procedures are conducted to identify the right set of hyperparameters (i.e., model settings). The ideal model parameters are

used to train 100 different ANN models, where the colocated data are randomly shuffled and split into $70\%$ training and $20\%$ validation data (referenced to the size of the original data set). Three binary classification metrics are calculated for every model run to evaluate the cloud classification performance for unseen data: the accuracy ($Ac$), F1 Score ($F1$), and Matthew's correlation coefficient ($Mcc$). Similarly, the Pearson product-moment correlation coefficient ($r$) and root-mean-square deviation (RMSD) provide the means to evaluate the performance of the $p_{\mathrm{CT}}$-models. Average values and standard deviations from each

set of 100 different model runs are $Ac = 0.934 \pm 0.001$, $F1 = 0.937 \pm 0.001$, $Mcc = 0.868 \pm 0.003$, $r = 0.819 \pm 0.001$, and RMSD$= 80.268 \pm 0.160\,\mathrm{hPa}$. The high statistical scores and low variability in the results illustrate that the two ANN algorithms yield reliable cloud classifications and $p_{\mathrm{CT}}$ estimates for previously unseen observations.

It is important to note that the predicted cloud parameters do not represent the true atmospheric state. Since each ANN was trained on the colocated MODIS targets, it follows that they, at best, will replicate the respective MODIS results. The MODIS

retrievals, however, are characterized by their own uncertainties and biases, which are subsequently learned and reproduced by the derived models. This means that analyses of ANN performance in this study only provide an evaluation of how well each model can replicate the colocated MODIS retrievals.

A comparison with the current v4.2x status flags reveals that, for both the validation and test data sets, the new ANN results provide a significant improvement in cloud classification. The ANN algorithm correctly identifies $> 93\%$ of cloudy profiles,

while less than $6\%$ of the clear profiles are falsely flagged. In contrast, the current v4.2x flag detects only $\approx 16\%$ of cloudy profiles, and even though it is designed to identify sufficiently opaque clouds, it only correctly classifies $< 27\%$ of cloudy profiles with $Q_{\mathrm{T}} > 1,000\,\mathrm{g\,m^{-2}}$. The fraction of falsely flagged clear profiles is comparable to the ANN results. Apart from a reduced ability to detect clouds over regions with generally low cloud cover, no significant dependence on geolocation is observed, indicating that the ANN flag yields reliable classification results on a global scale. A global cloud cover map for data

collected between 2005 and 2019 is presented, generated solely from MLS-sampled $T_{\mathrm{B}}$ and the determined ANN weights. Typically observed cloud patterns and derived $C$ agree reasonably well with MODIS results. Moreover, detailed examination of four example scenes from the North American and Asian summer monsoon regions reveals that the ANN can reliably

identify diverse cloud fields, including those characterized by low-level clouds and low $Q_\text{T}$. Together with the consistently large statistical agreement, these global and regional examples of successful cloud detection illustrate that the predefined cloudiness conditions (following thresholds for $C$, $p_\text{CT}$, and $Q_\text{T}$) are reasonable. Moreover, the uncertainties arising from associating MLS observations in the limb with nadir MODIS images do not seem to substantially impact the reliability of the ANN algorithm.

Similarly, the second ANN setup is able to reliably estimate the MODIS-retrieved $p_\text{CT}$ for profiles in the validation and test data set, with $r > 0.82$ and RMSD $< 80\,\text{hPa}$. It is shown that more than $66\%$ of $p_\text{CT}$ predictions are within $50\,\text{hPa}$ of the MODIS results. Derived global maps of average ANN-predicted $p_\text{CT}$ can reproduce observed patterns in the MODIS retrievals. In particular, this model is able to correctly identify $> 85\%$ of profiles with $p_\text{CT} < 400\,\text{hPa}$ in the $-60°$ to $+60°$ latitude range. Conversely, the current v4.2x algorithm correctly flags only $\approx 9\%$ of such profiles as cloudy.

This new cloud classification scheme, which will be included in future versions of the MLS dataset, provides the means to reliably identify profiles with potential mid- to high-level cloud influence. Note that MLS radiances are not affected by the change from v4.2x to v5.0x. As mentioned in the introduction, this new algorithm will facilitate future research on reducing uncertainties in the retrieval of atmospheric constituents in the presence of clouds. Moreover, studies on convective moistening of the lowermost stratosphere, as well cloud scavenging of atmospheric pollutants, will benefit from these new capabilities.

*Data availability.* MLS brightness temperatures and L2GP data, including status flags, are available at https://mls.jpl.nasa.gov. Aqua-MODIS data are obtained from the LAADS-DAAC at https://ladsweb.modaps.eosdis.nasa.gov/search/order/1/MODIS:Aqua

**Table A1.** Details about the colocated MLS-MODIS data set, which contains observations from thirteen random Julian days (d01-d13) for each year over 2005–2020.

| | 2005 | 2006 | 2007 | 2008 | 2009 | 2010 | 2011 | 2012 | 2013 | 2014 | 2015 | 2016 | 2017 | 2018 | 2019 | 2020 |
|---|---|---|---|---|---|---|---|---|---|---|---|---|---|---|---|---|
| d01 | 001 | 005 | 006 | 018 | 031 | 021 | 010 | 004 | 014 | 002 | 015 | 010 | 021 | 008 | 010 | 008 |
| d02 | 034 | 041 | 041 | 055 | 055 | 041 | 040 | 036 | 037 | 052 | 016 | 031 | 053 | 043 | 041 | 048 |
| d03 | 064 | 045 | 066 | 064 | 056 | 064 | 076 | 061 | 079 | 065 | 033 | 044 | 069 | 075 | 083 | 087 |
| d04 | 117 | 066 | 092 | 068 | 079 | 092 | 077 | 101 | 113 | 101 | 064 | 076 | 106 | 109 | 109 | 181 |
| d05 | 121 | 098 | 134 | 076 | 092 | 126 | 112 | 134 | 121 | 139 | 116 | 110 | 107 | 124 | 139 | 192 |
| d06 | 122 | 099 | 173 | 119 | 145 | 164 | 142 | 169 | 154 | 167 | 124 | 140 | 143 | 163 | 181 | 196 |
| d07 | 158 | 140 | 197 | 122 | 164 | 201 | 177 | 187 | 206 | 202 | 159 | 172 | 169 | 183 | 189 | 214 |
| d08 | 206 | 161 | 230 | 158 | 212 | 240 | 185 | 224 | 238 | 238 | 186 | 212 | 205 | 223 | 239 | 215 |
| d09 | 223 | 194 | 265 | 197 | 221 | 264 | 224 | 225 | 273 | 255 | 216 | 213 | 243 | 269 | 256 | 247 |
| d10 | 244 | 242 | 302 | 198 | 251 | 275 | 261 | 246 | 274 | 256 | 248 | 233 | 270 | 280 | 257 | 251 |
| d11 | 286 | 259 | 322 | 224 | 286 | 276 | 293 | 300 | 313 | 290 | 284 | 266 | 283 | 281 | 286 | 258 |
| d12 | 319 | 301 | 323 | 292 | 314 | 308 | 317 | 332 | 337 | 330 | 309 | 299 | 332 | 324 | 308 | 284 |
| d13 | 349 | 323 | 360 | 328 | 353 | 356 | 359 | 350 | 338 | 354 | 342 | 323 | 358 | 358 | 341 | 310 |

## Appendix A: Days in the MLS-MODIS data set

The following table lists the days included in the colocated MLS-MODIS data set. Days were semi-randomly chosen to ensure that each month is represented equally and only complete measurement days (i.e., to technical issues with the instruments) are included.

*Author contributions.* All authors have shaped the concept of this study and refined the approach during extensive discussions. FW carried
out the data analysis and wrote the initial draft of the manuscript, which was subsequently refined by all authors.

*Competing interests.* The authors declare that they have no conflict of interest.

*Acknowledgements.* ©2020. California Institute of Technology. Government sponsorship acknowledged. The research was carried out at the Jet Propulsion Laboratory, California Institute of Technology, under a contract with the National Aeronautics and Space Administration (80NM0018D0004).

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

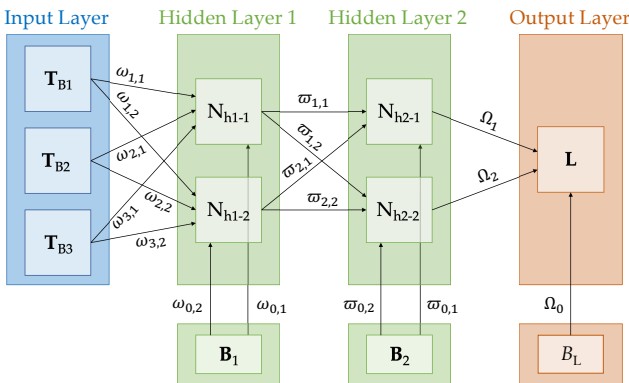

**Figure 1.** Simplified sketch of the algorithm setup, including three vectors in the input layer (blue) that contain MLS brightness temperatures ($\mathbf{T}_{Bi}$; $i=1-3$), two hidden layers (green) with two neurons ($N_{h1-k}$ and $N_{h2-k}$; $k=1-2$) and one "bias" node each ($\mathbf{B}_k$; $k=1-2$), and an output layer (orange) with the labels vector ($\mathbf{L}$) and one "bias" node ($\mathbf{B}_L$). Also shown are the input weights ($\omega_{i,k}$; $i=0-3$, $k=1-2$), connecting weights ($\varpi_{k,l}$; $k=0-2$, $l=1-2$), and output weights ($\Omega_l$; $l=0-2$) that connect the input variables to the neurons in the first hidden layer, the neurons from the two hidden layers, and the neurons from the second hidden layer to the labels vector, respectively.

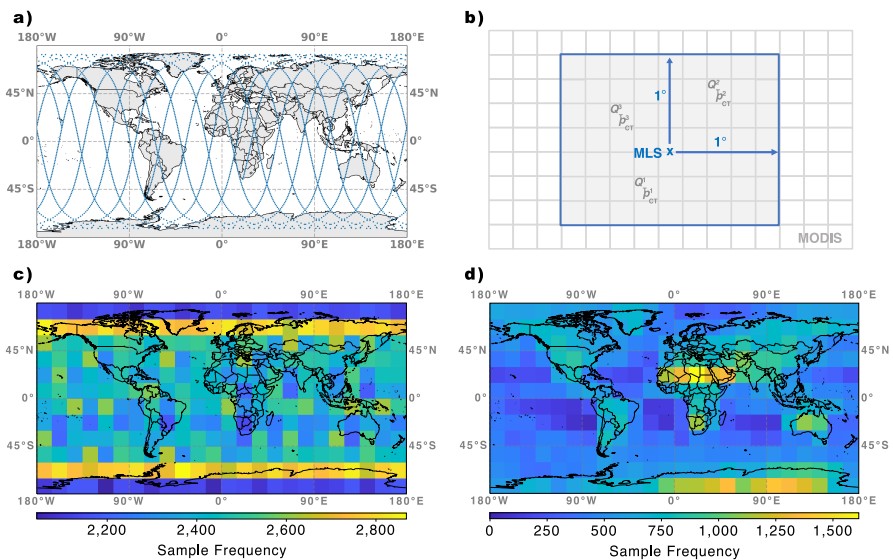

**Figure 2.** (a) Example MLS orbit on 19 May 2019. (b) Illustration of the colocation of MLS and MODIS data. (c) Global map of sample frequencies for the colocated MLS-MODIS data set used in this study. (d) Same as (c), but showing the sample frequencies of observed clear and cloudy profiles, following the definitions in section 3.2.

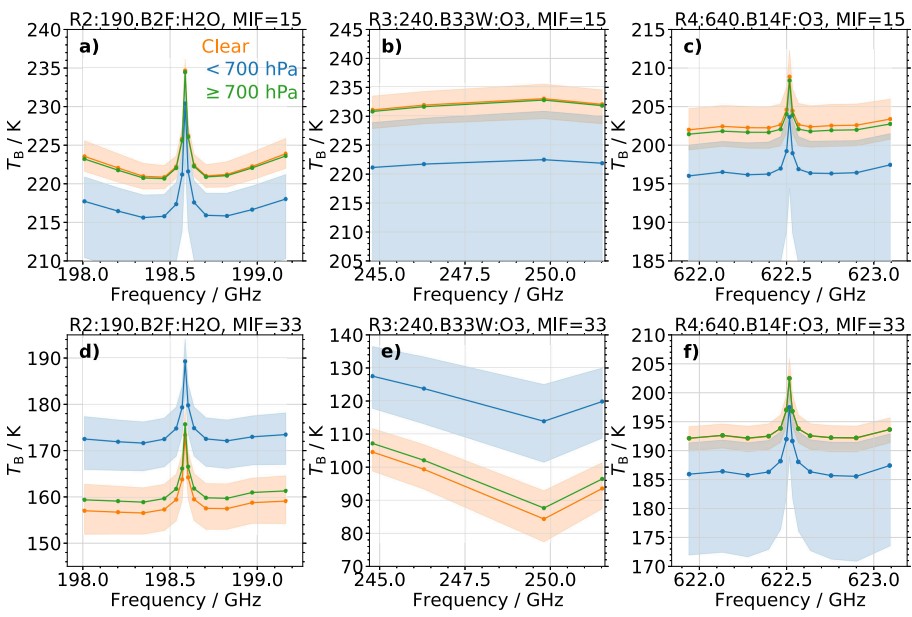

**Figure 3.** (a) Statistic of the brightness temperature ($T_B$) from MLS observations sampled in band 2 of receiver 2 at minor frame (MIF) 15 (at an altitude of $\approx 4.5\,\mathrm{km}$) in the latitudinal range of $-30°$ to $+30°$ as a function of frequency. The orange, blue, and green curves show the median $T_B$ associated with clear sky conditions, clouds with a cloud top pressure $p_{CT} < 700\,\mathrm{hPa}$, and clouds with $p_{CT} \geq 700\,\mathrm{hPa}$, respectively. The shaded orange and blue areas indicate the interquartile range of the respective $T_B$ (omitted for low clouds to enhance legibility). Samples are provided by the colocated MLS-MODIS data set. (b) Same as (a), but for band 33 of radiometer 3. (c) Same as (a), but for band 14 of radiometer 4. (d)–(f) Same as (a)–(c), but at MIF=33 (at an altitude of $\approx 12\,\mathrm{km}$).

**Table 2.** Details of the input variables for the ANN algorithm, which consist of MLS brightness temperature observations in 10 different bands from 4 radiometers. Besides the official radiometer and band designations, the local oscillator (LO) and primary species of interest in the respective band are given, as well as the ranges of minor frames (MIFs) and channels used as input for the ANN.

| Spectrometer | Band | LO (GHz) | Species | MIF | Channel |
|---|---|---|---|---|---|
| R1A | B1F | 118 | $p_{tan}$ | [7, 10, 13, …, 49] | [1, 3, 5 , …, 25] |
| R2 | B2F | 190 | $H_2O$ | [7, 10, 13, …, 49] | [1, 3, 5 , …, 25] |
| R2 | B3F | 190 | $N_2O$ | [7, 10, 13, …, 49] | [1, 3, 5 , …, 25] |
| R2 | B6F | 190 | $O_3$ | [7, 10, 13, …, 49] | [1, 3, 5 , …, 25] |
| R3 | B7F | 240 | $O_3$ | [7, 10, 13, …, 49] | [1, 3, 5 , …, 25] |
| R3 | B8F | 240 | $p_{tan}$ | [7, 10, 13, …, 49] | [1, 3, 5 , …, 25] |
| R3 | B33W | 240 | $O_3$ | [7, 10, 13, …, 49] | [1, 2, 3, 4] |
| R4 | B10F | 640 | ClO | [7, 10, 13, …, 49] | [1, 3, 5 , …, 25] |
| R4 | B14F | 640 | $O_3$ | [7, 10, 13, …, 49] | [1, 3, 5 , …, 25] |
| R4 | B28M | 640 | $HO_2$ | [7, 10, 13, …, 49] | [1, 3, 5 , …, 11] |

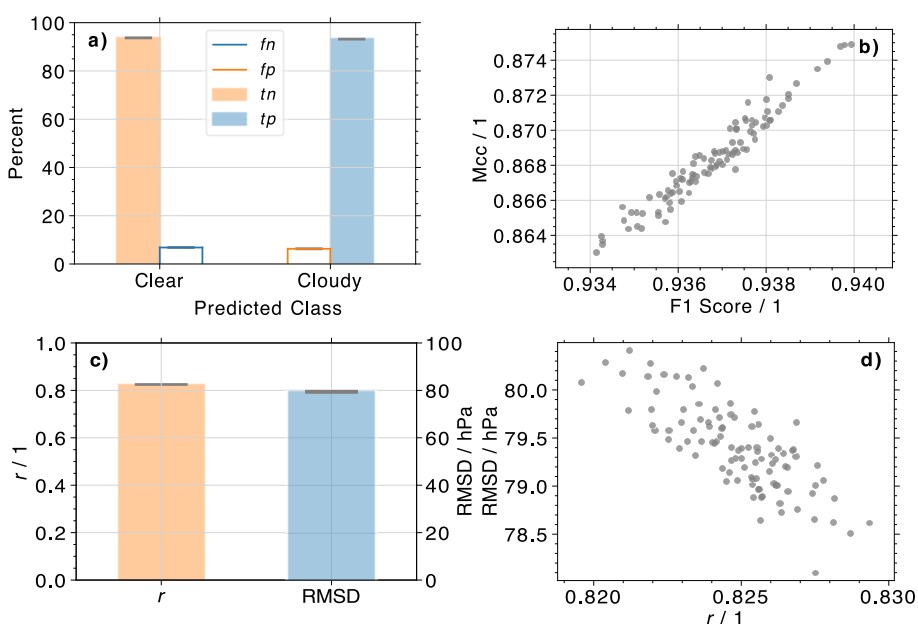

**Figure 4.** (a) Histograms of cloud classifications from the ANN algorithm for 100 random combinations of training and validation data sets. Orange and blue shading depicts the percent of correctly predicted clear (i.e., true negatives, $tn$) and cloudy (i.e., true positives, $tp$) labels for actually observed clear and cloudy profiles, respectively. Orange and blue lines depict the percent of falsely predicted cloudy (i.e., false positives, $fp$) and clear (i.e., false negatives, $fn$) labels for actually observed clear and cloudy profiles, respectively. The vertical extent of the gray horizontal bars on top of each histogram indicates the standard deviation derived from all 100 predictions (the horizontal extent is arbitrary). (b) Scatter plot of Matthews correlation coefficient ($Mcc$) and F1 score for the same 100 random combinations of training and validation data sets shown in (a). (c) Similar to (a), but showing histograms of derived Pearson product-moment correlation coefficient ($r$) and root-mean-square deviation (RMSD) from the ANN cloud top pressure algorithm. (d) Similar to (b), but showing the relationship between $r$ and RMSD.

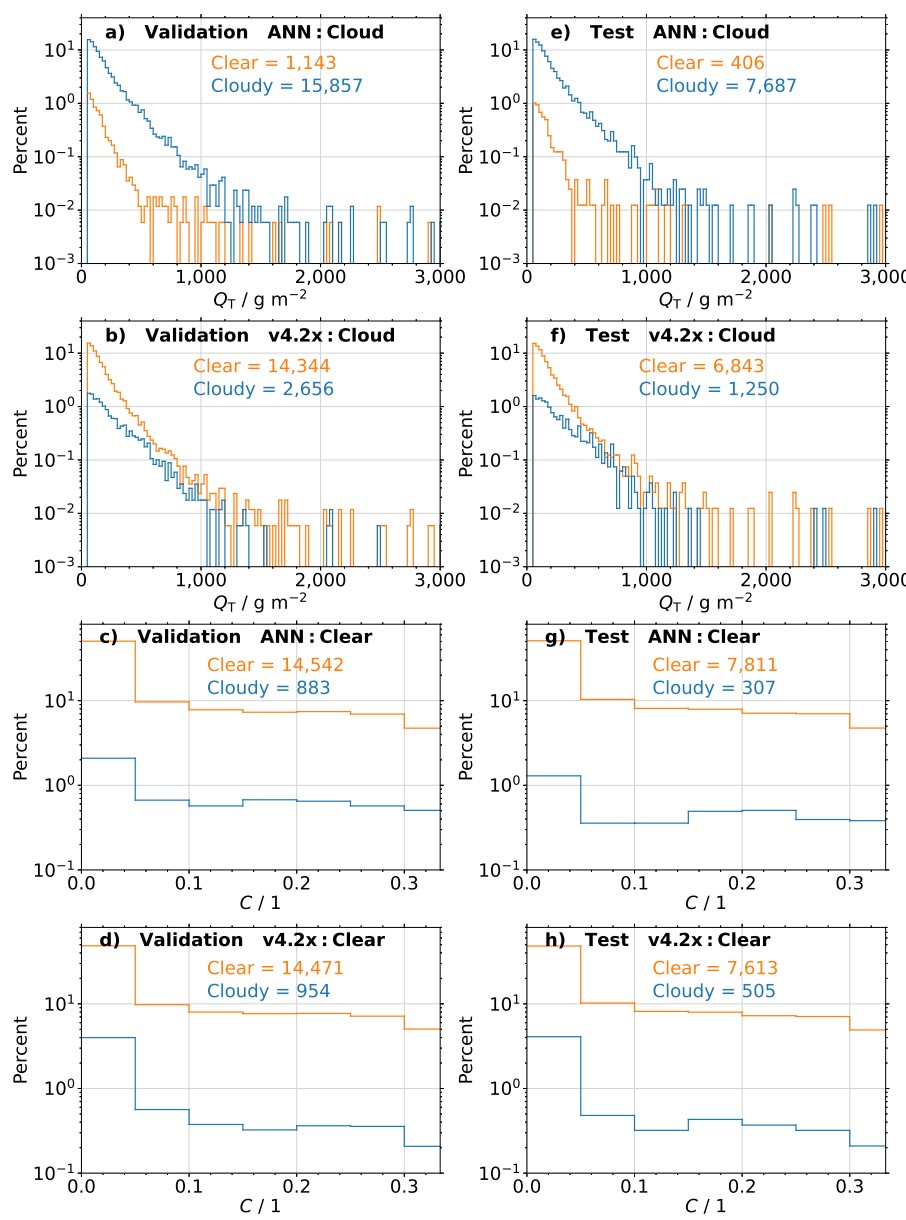

**Figure 5.** (a) Histograms of cloud classifications from the new ANN-based cloud flag for actually observed cloudy profiles as a function of total water path ($Q_T$). Only profiles from the validation data set are considered. Orange and blue colors depict the distributions of predicted clear and cloudy labels, respectively. The number of clear and cloudy predictions is also given. (b) Same as (a), but for classifications from the current v4.2x cloud flag. (c)-(d) Similar to (a)-(b), but for actually observed clear profiles as a function of cloud cover ($C$). (e)-(h) Same as (a)-(d), but for profiles from the test data set.

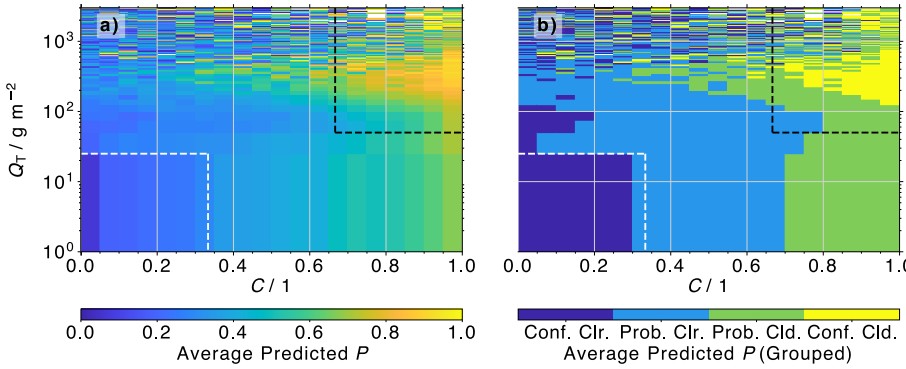

**Figure 6.** (a) Average probability of cloudiness ($P$) predicted by the ANN as a function of $C$ and $Q_T$. No restrictions on cloud top pressure ($p_{CT}$) are imposed. (b) Same as (a), but $P$ is grouped into four classes: confidently clear ("Conf. Clr."; $P < 0.25$), probably clear ("Prob. Clr."; $0.25 \leq P < 0.5$), probably cloudy ("Prob. Cld."; $0.5 \leq P < 0.75$), and confidently cloudy ("Conf. Cld."; $P \geq 0.75$).

**Table 3.** Binary classification statistics for the new ANN algorithm, as well as the classification provided by the current MLS v4.2x status flag. Prescribed labels (i.e., clear sky or cloudy) are provided by the standard definitions presented in section 3.2; statistics are given for both the validation and test data sets. Results for two modified definitions based on looser thresholds are also given; here statistics are based on all profiles in the MLS-MODIS data set (minus the training data). The fraction of true positives and negatives ($tp$ and $tn$), as well as false positives and negatives ($fp$ and $fn$), are given. Finally, three measures for the evaluation of binary statistics are listed: the accuracy ($Ac$), the F1 score ($F1$), and the Matthews correlation coefficient ($Mcc$).

|  | $tp$ | $tn$ | $fp$ | $fn$ | $Ac$ | $F1$ | $Mcc$ |
|---|---|---|---|---|---|---|---|
| ANN (validation) | 0.93 | 0.94 | 0.06 | 0.07 | 0.94 | 0.94 | 0.87 |
| v4.2x (validation) | 0.16 | 0.94 | 0.06 | 0.84 | 0.53 | 0.26 | 0.15 |
| ANN (test) | 0.95 | 0.96 | 0.04 | 0.05 | 0.96 | 0.96 | 0.91 |
| v4.2x (test) | 0.15 | 0.94 | 0.06 | 0.85 | 0.55 | 0.25 | 0.15 |
| ANN (modified, all $p_{CT}$) | 0.58 | 0.91 | 0.09 | 0.42 | 0.65 | 0.73 | 0.41 |
| v4.2x (modified, all $p_{CT}$) | 0.05 | 0.95 | 0.05 | 0.95 | 0.24 | 0.09 | 0.00 |
| ANN (modified, $p_{CT} < 700\,\mathrm{hPa}$) | 0.89 | 0.91 | 0.09 | 0.11 | 0.90 | 0.90 | 0.81 |
| v4.2x (modified, $p_{CT} < 700\,\mathrm{hPa}$) | 0.13 | 0.95 | 0.05 | 0.87 | 0.54 | 0.22 | 0.13 |

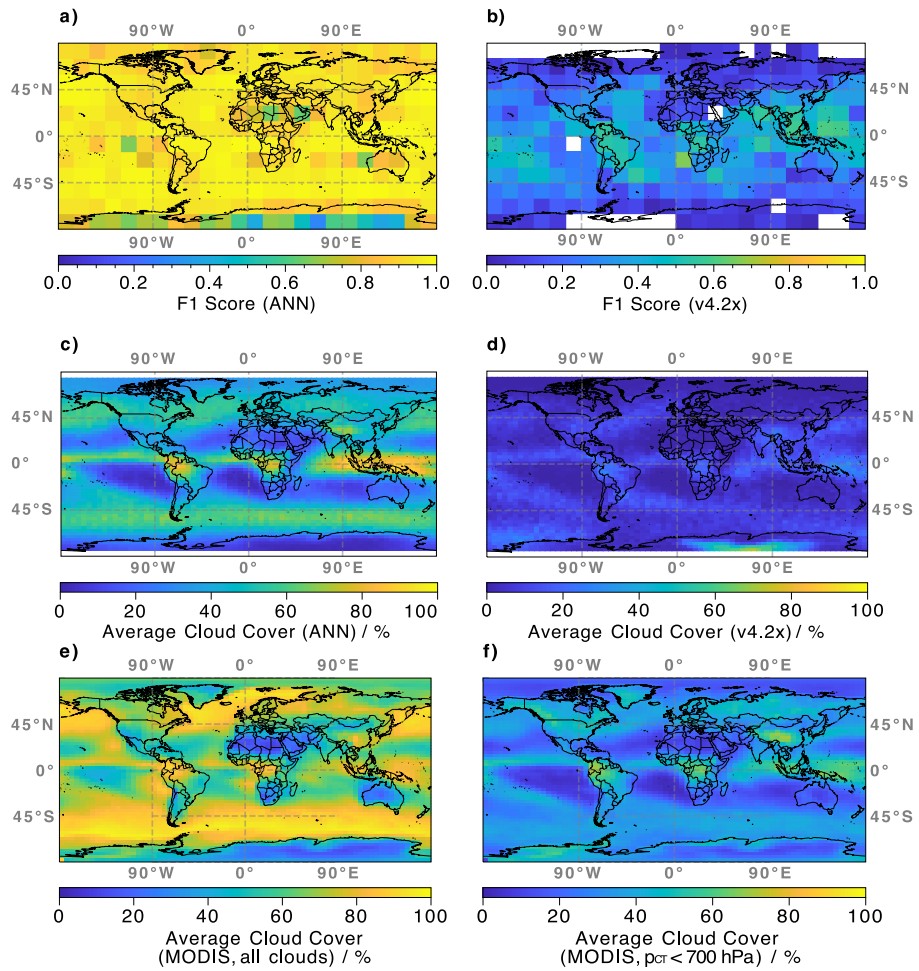

**Figure 7.** (a) Latitudinal and longitudinal dependence of the performance of the ANN algorithm, determined by the F1 score for binary classifications. Observations and actual cloudiness flags are provided by the colocated MLS-MODIS data set; only profiles from the validation and training data set are considered. (b) Same as (a), but for the current v4.2x cloud flag. (c) Average global cloud cover derived from MLS brightness temperature observations and the weights determined from the trained ANN. All MLS observations sampled between 2015 and 2019 are represented. (d) Same as (c), but for the current v4.2x cloud flag. (e) Same as (c), but from Aqua-MODIS observations sampled in 2019. (f) Same as (e), but with retrieved cloud top pressure $< 700\,\mathrm{hPa}$.

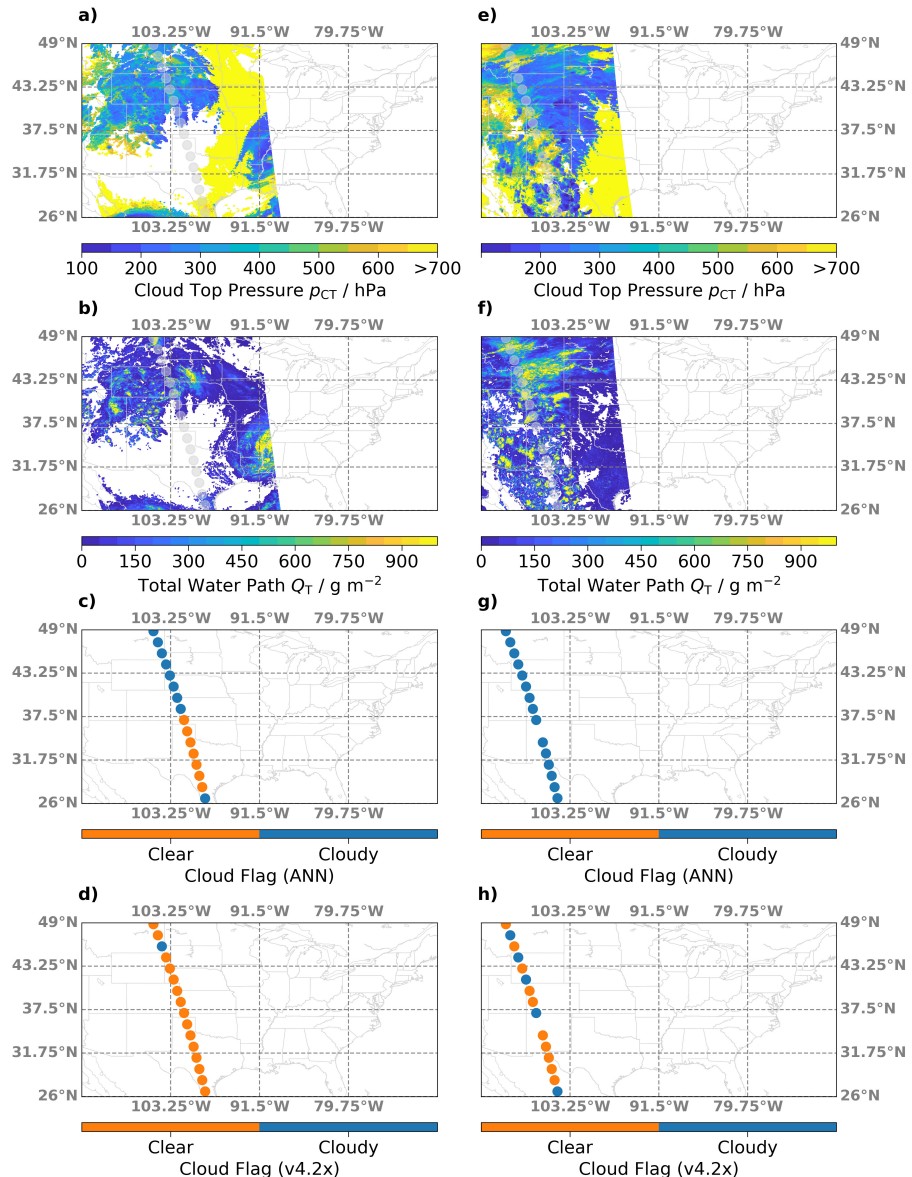

**Figure 8.** (a) Map of cloud top pressure ($p_{CT}$) retrieved from MODIS observations on 31 August 2017 over North America. Transparent circles indicate the MLS orbit. (b) Similar to (a), but for the total water path ($Q_t$). (c) Clear (orange) and cloudy (blue) profiles as determined from the new ANN algorithm. (d) Same as (c), but determined from the current v4.2x status flags. (e)-(h) Same as (a)-(d), but for MLS and MODIS observations on 5 July 2015.

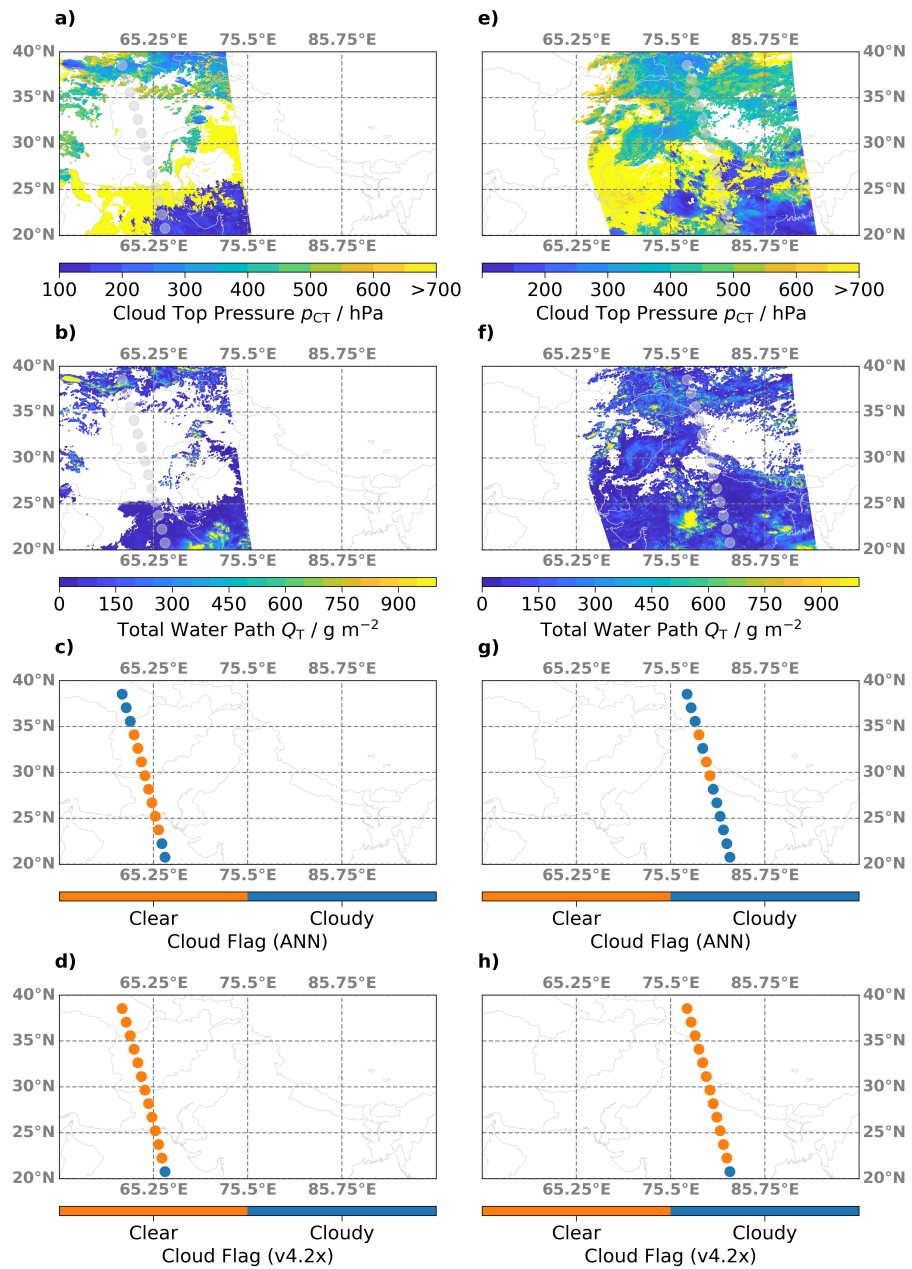

**Figure 9.** Similar to Figure 8, but for MLS and MODIS observations on (a)-(d) 28 June 2019 and (e)-(h) 5 July 2018, respectively, over South Asia. These scenes were captured over the Asian summer monsoon region.

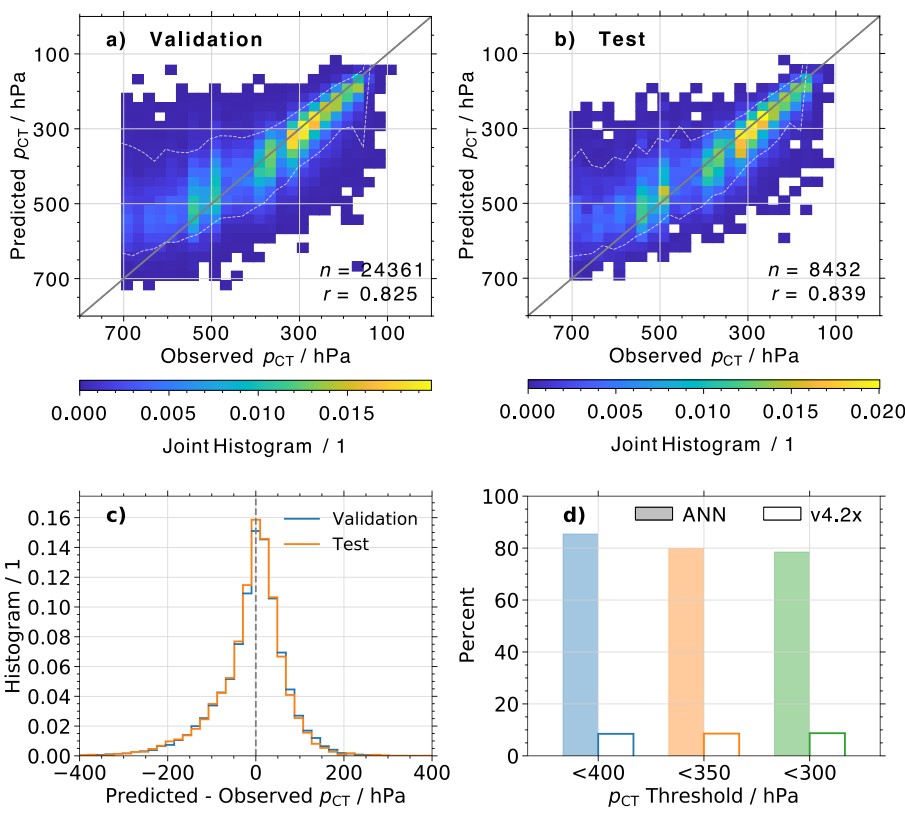

**Figure 10.** (a) Normalized joint histograms of true and predicted cloud top pressure ($p_{CT}$). Data are from the validation data set. (b) Same as (a), but for the training data set. (c) Histograms of the difference between predicted and observed $p_{CT}$, for profiles in the validation (blue) and test (orange) data set. (d) Percent of observed $p_{CT} < 400, 350, 300$ hPa that were successfully detected by the ANN (color-filled bars) and flagged by the v4.2x algorithm (transparent bars). Data are from both the validation and test data sets.

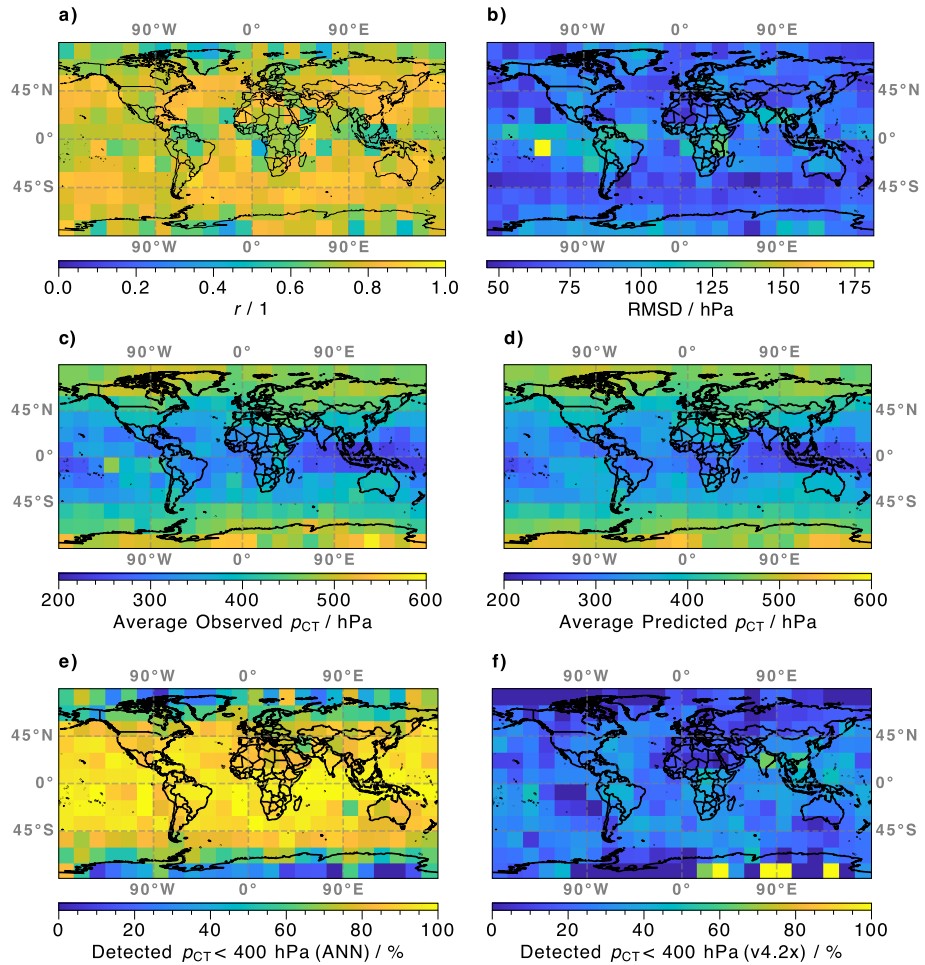

**Figure 11.** (a) Map of derived Pearson product-moment correlation coefficient ($r$) between MODIS-retrieved and ANN-predicted cloud top pressure ($p_{CT}$). Observations are provided by the colocated MLS-MODIS data set; only profiles from the validation and training data set are considered. (b) Similar to (a), but showing the root-mean-square deviation (RMSD). (c) Similar to (a), but showing the average MODIS-retrieved $p_{CT}$. (d) Same as (c), but showing the ANN predictions. (e) Similar to (a), but showing the percent of observed $p_{CT} < 400$ hPa that are successfully detected by the ANN. (f) Same as (e), but showing the percent where the v4.2x algorithm detected a cloud.

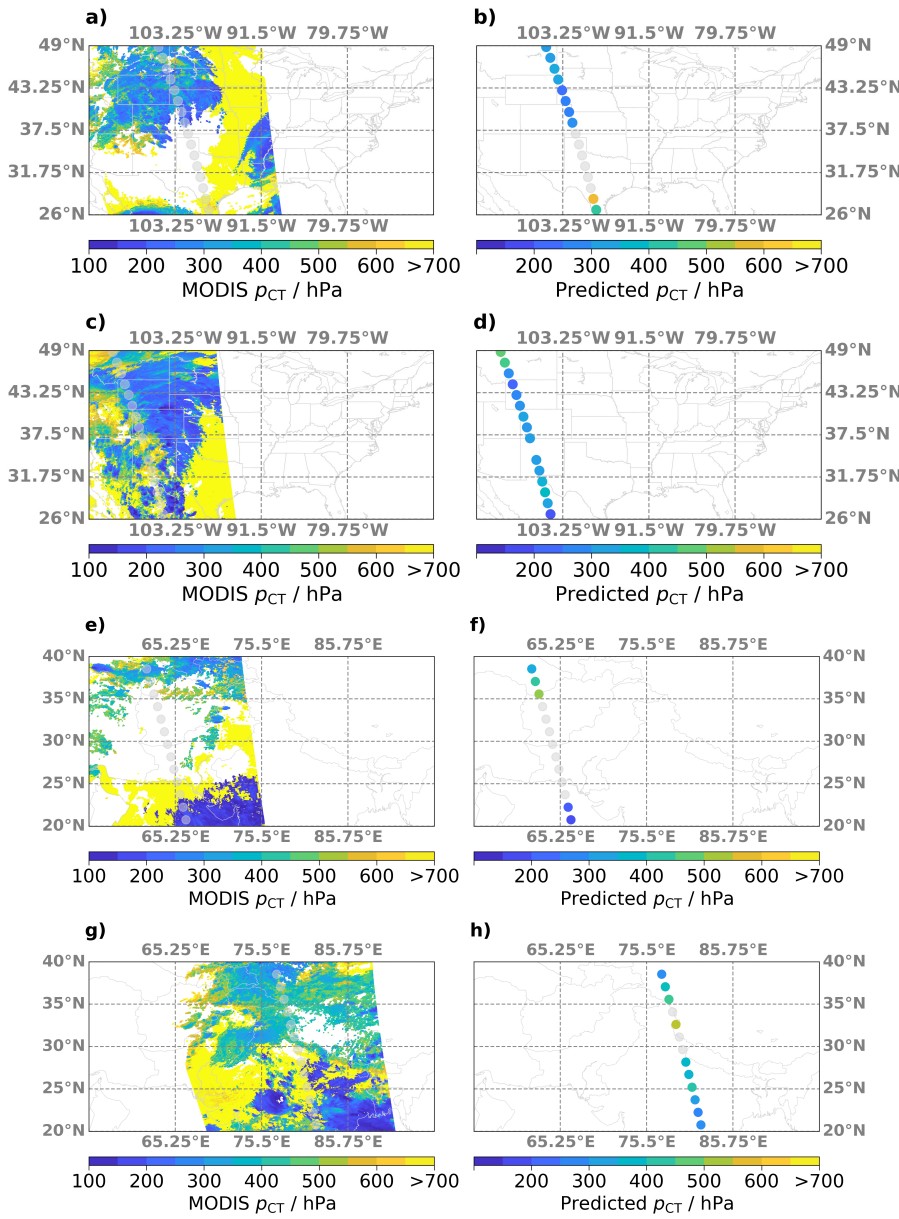

**Figure 12.** (a) Map of cloud top pressure ($p_{CT}$) retrieved from MODIS observations on 31 August 2017 over North America. Transparent circles indicate the MLS orbit. (b) Same as (a), but for the predicted $p_{CT}$ based on the ANN algorithm. (c)-(d) Same as (a)-(b), but for MLS and MODIS observations on 5 July 2015. (e)-(f) Similar to (a)-(b), but for MLS and MODIS observations on 28 June 2019. (g)-(h) Same as (a)-(b), but for MLS and MODIS observations on 5 July 2018.