# Peer review of "Improved cloud detection for the Aura Microwave Limb Sounder: Training an artificial neural network on colocated MLS and Aqua-MODIS data"

_Atmospheric Measurement Techniques, 2021_

## Referee Comment (RC1)

Review of the paper "Improved cloud detection for the Aura Microwave Limb Sounder: Training an artificial neural network on colocated MLS and Aqua-MODIS data" by Frank Werner et al.

General comments

This paper nicely illustrates that the implementation of machine learning to MLS cloud classification leads to an impressive improvement in MLS cloud detection, compared to current operational techniques.

The paper is concise, well written, and discusses well-selected calculations. The discussion of both global statistics, and individual cases, is very appealing. The Summary and Conclusions section is very well written.

The discussion of the machine learning methodology is very concise, but could benefit by briefly defining some of the machine learning terms which may not be familiar to the atmospheric science research community.

The paper should be published following very minor suggested changes.

Specific comments

The use of machine learning techniques and terminology is likely unfamiliar to many in the atmospheric sciences. There are several places in the text in which a few additional words / sentences could help the reader understand better what is being done by the authors. There are some terms which need to be defined. Please discuss, for example, what is meant by "feedforward" on line 121. Other terms that should be defined (briefly discussed) are "imbalanced classes", "learning rate", "Nesterov momentum value", and "weight *decay*".

Technical comments

Line 21 the phrase "cloud amount" is vague. Please be more specific.

Line 46, add commas, revising to e.g. "radiances, from lower in the atmosphere, and smaller downwelling radiances from above, into the MLS raypath" to improve readability. In my first reading of the sentence I had a hard time making sense of the sentence.

Line 55, what is meant by "discount them" ?

Line 89, please specify Figures in Waters et al 2006 or other papers that illustrate the spectral sampling details of the AURA MLS experiment, so the reader can obtain a fuller understanding of the MLS experiment.

Line 130. It would be helpful to point out that Figure 1 is presented for illustrative purposes, since line 253 later points out that each hidden layer has 851 neurons (instead of 2 neurons).

"Figure 1 illustrates the general setup of a simplified multilayer perceptron that contains four layers, and is instructional. The full model setup is discussed in Section 3.4"

Line 168. Is the MLS aggregation at 1°x1° because the MLS data sampling is (line 100) near 165 km?

Line 173 are the 5,000 samples MODIS, MLS, or MODIS-MLS samples?

Line 262. Approximately how many epochs are calculated?

Line318 clarify what is meant by "classification going forward".

Line 549. If the current MLS data version is V5, why not include the new ANN capability in the V5 product instead of "future versions of the v4.2x" product?

Standard Criteria

1. Does the paper address relevant scientific questions within the scope of AMT?  Yes

2. Does the paper present novel concepts, ideas, tools, or data? Yes. The discussion of machine learning techniques is relatively new to the atmospheric sciences, and very appropriate.

3. Are substantial conclusions reached? Yes, the machine learning technique makes an impressive improvement in MLS cloud detection.

4. Are the scientific methods and assumptions valid and clearly outlined? Yes

5. Are the results sufficient to support the interpretations and conclusions? Yes

6. Is the description of experiments and calculations sufficiently complete and precise to allow their reproduction by fellow scientists (traceability of results)? Yes

7. Do the authors give proper credit to related work and clearly indicate their own new/original contribution? Yes

8. Does the title clearly reflect the contents of the paper? Yes

9. Does the abstract provide a concise and complete summary? Yes

10. Is the overall presentation well structured and clear? Yes

11. Is the language fluent and precise? Yes (with only one or two exceptions, noted above)

12. Are mathematical formulae, symbols, abbreviations, and units correctly defined and used? Yes

13. Should any parts of the paper (text, formulae, figures, tables) be clarified, reduced, combined, or eliminated? A few clarifications, noted above, are suggested.

14. Are the number and quality of references appropriate? Yes (with perhaps one place to add specific reference to figures that illustrate the AURA MLS spectral territory).

15. Is the amount and quality of supplementary material appropriate? Supplementary material is not included in this paper

---

## Author Comment (AC1)

We'd like to thank the editor for handling our manuscript, as well as reviewer #1 for reading our manuscript and providing numerous helpful suggestions for improvement.

We have carefully read through all the comments and questions and revised the manuscript accordingly. Please find our point-by-point response to reviewer #1 below. Here, the reviewer's general remarks, as well as the specific questions/comments, are formatted to be left-aligned text in bold font. Our responses are indented and formatted in regular font.

Here is a summary of the major changes in the revised manuscript:

1) We defined an independent test data set that is not included in the determination of hyperparameters or the training of the ANNs. The new splits are 70%/20%/10% for the training/validation/test data set. We also excluded the training data set from the evaluation of the model performance. This means that, e.g., the histograms in Figure 5 or the global maps in Figure 6 are generated by profiles the ANN has not been trained on. This allows for a fairer evaluation of ANN performance.

2) Since we removed profiles for the test data set and introduced new splits between training and validation data, we needed to repeat the k-fold cross validation and training of the ANNs. This turned out to be a necessary step, as we were able to fix three bugs in our algorithm setup: (i) We had not considered the number of hidden layers to be a hyperparameter. Tests revealed that models with only one hidden layer slightly outperformed those with two layers for the cloud classification scheme (the cloud top pressure models still use two layers). (ii) We had shuffled the training and validation data twice. While this had (obviously) no effect on training performance, it affected the correct recording of the respective profile indices. In other words, we did not correctly track the profiles in the training and validation data sets. This, in turn, means that the validation statistics presented in Figures 4 and 10 were inaccurate, as the presented "validation data" was actually comprised of random profiles from both the training and validation data set. Note that in the original manuscript version, model performance was evaluated for the combined training and validation data set (e.g., Figures 5 and 7), which means this mistake had no effect (i.e., the evaluation was based on a combined data set). (iii) The wrong control file provided the cloud top pressure model in the original manuscript version. That specific model had no weight decay (i.e., the model could learn training data very well, to the detriment of generalization) and early-stopping was turned off. This, together with the wrong recording of training and validation indices, resulted in the unrealistic correlation coefficients of 0.99. The model in the revised manuscript exhibits a much more reasonable correlation coefficient of 0.82.

3) We added more detailed explanations of machine learning terminology and descriptions of the considered hyperparameters.

4) We replaced one of the example scenes over South East Asia. In the original manuscript, the two scenes in Figure 9 looked very similar. Instead, we decided to present a more complex cloud field, which nicely illustrates the performance of the cloud classification model, while highlighting instances where the cloud top pressure prediction struggles.

5) We extended the analysis of the cloud top pressure ANN performance considerably. That section now includes additional statistical analysis of the difference between predictions and observations, as well as the model's ability to detect clouds <400, 350, and 300 hPa.

We also added global maps of the model performance, as well as comparisons between MODIS, ANN, and v4.2x data (similar to the cloud classification analysis). Example maps now contain the same scenes as for the cloud classification part.

**General comments**

**This paper nicely illustrates that the implementation of machine learning to MLS cloud classification leads to an impressive improvement in MLS cloud detection, compared to current operational techniques.**

**The paper is concise, well written, and discusses well-selected calculations. The discussion of both global statistics, and individual cases, is very appealing. The Summary and Conclusions section is very well written.**

**The discussion of the machine learning methodology is very concise, but could benefit by briefly defining some of the machine learning terms which may not be familiar to the atmospheric science research community.**

**Specific comments**

**The use of machine learning techniques and terminology is likely unfamiliar to many in the atmospheric sciences. There are several places in the text in which a few additional words / sentences could help the reader understand better what is being done by the authors. There are some terms which need to be defined. Please discuss, for example, what is meant by "feedforward" on line 121. Other terms that should be defined (briefly discussed) are "imbalanced classes", "learning rate", "Nesterov momentum value", and "weight *decay*".**
We added the following descriptions to the manuscript.

"Here, we constructed and trained a multilayer perceptron, which is a subcategory of feedforward ANNs that sequentially connects neurons between different layers. In a feedforward ANN information only gets propagated forward through the different model layers and is not directed back to affect previous layers."

And:
"Generally, *F1* assigns more relevance to false predictions and is more suitable for imbalanced classes, where the respective data sizes vary significantly."

And:
"The hyperparameters to be determined are (i) the number of hidden layers, (ii) the number of neurons per hidden layer, (iii) the optimizer for the cloud classification, (iv) the mini-batch size, (v) the learning rate, and (vi) the value for the weight decay (i.e., the L2 regularization parameter). The number of hidden layers and neurons impact the complexity of the model. The choice of optimizer controls how fast and accurately the minimum of the loss function in Eq. (8) is determined, based on different feature sets and

minimization techniques. During each iteration the model computes an error gradient and updates the model weights accordingly. Instead of determining the error gradient from the full training data set, our models only use a random subset of the training data (called a mini-batch) during each iteration. This not only speeds up the training process, but also introduces noise in the estimates of the error gradient, which improves generalization of the models. The learning rate controls how quickly the weights are updated along the error gradient. Thus, the size of the learning rate affects the speed of convergence (higher is better) and ability to detect local minima in the loss function (lower is better). Meanwhile, L2 regularization is one method to specify the regularization term $R$ in Eq. (8), where the sum of the squared weights is multiplied with the L2 parameter:

$$R = L2 \cdot \sum \omega^2 + \varpi^2 + \Omega^2 \quad (9)$$

Note that for clarity we omitted the indices for the weights in Eq (8). The amount of regularization is directly proportional to the value of the L2 weight decay parameter. Regularization usually improves generalization of the models. More information about ANN hyperparameters and their impact on the reliability of model predictions can be found in, e.g., Reed and Marks (1999) and Goodfellow et al. (2016)."

Note that due to changes in setting up the models, as well as the performance evaluation, we found that the Adam optimizer slightly outperforms the stochastic gradient one. We changed the description accordingly.

**Technical comments**

**Line 21 the phrase "cloud amount" is vague. Please be more specific.**
    We changed the wording to "cloud cover".

**Line 46, add commas, revising to e.g. "radiances, from lower in the atmosphere, and smaller downwelling radiances from above, into the MLS raypath" to improve readability. In my first reading of the sentence I had a hard time making sense of the sentence.**
    We changed the sentence following the reviewer's recommendation: "a mix of large upwelling radiances, from lower in the atmosphere, and smaller downwelling radiances, from above…".

**Line 55, what is meant by "discount them" ?**
    We meant to say that these radiances are discarded when the observation vector for the optimal estimation is constructed. We changed the wording to "discard".

**Line 89, please specify Figures in Waters et al 2006 or other papers that illustrate the spectral sampling details of the AURA MLS experiment, so the reader can obtain a fuller understanding of the MLS experiment.**
    We added "; see Table 4 in Waters et al. (2006) and Figure 2.1.1 in Livesey et al. (2020)." to the revised manuscript.

**Line 130. It would be helpful to point out that Figure 1 is presented for illustrative purposes, since line 253 later points out that each hidden layer has 851 neurons (instead of 2 neurons). "Figure 1 illustrates the general setup of a simplified multilayer perceptron that contains four layers, and is instructional. The full model setup is discussed in Section 3.4"**

> We changed the sentence as follows: "Figure 1 illustrates the general setup of a simplified multilayer perceptron that contains four layers, and is purely instructional. The complete model setup is more complex and is discussed in sections 3.2–3.4."

**Line 168. Is the MLS aggregation at 1°x1° because the MLS data sampling is (line 100) near 165 km?**

> We spent quite some time thinking about this detail. Indeed, the half and full distance between adjacent MLS profiles is close to 0.75° and 1.5°, respectively. At the same time, the typical horizontal scales of clouds that can potentially impact MLS observations (i.e., optically thick mid-level cloud fields and high-reaching cumulonimbus) are in the range of 50-200 km (Guillaume et al. 2018). This gives us a range of ~0.5°-2.0°.

> We tested the aggregation for different scales 0.5°-3.0° in increments of 0.5° to get an idea about the importance of the aggregation perimeter. We noticed no significant difference in performance for scales between 0.5° and 2.0° (variability in Matthew's Correlation Coefficient of <0.01). However, performance got gradually worse for 2.5° and 3.0°.

> In the end we decided on 1°x1°, which is (i) close to half of the distance between adjacent MLS profiles, and (ii) in the middle of the relevant horizontal cloud scales.

> We added some extra information to the manuscript at the end of the third paragraph of section 3.2: "Note that no significant decrease in classification performance is observed for varying aggregation scales between 0.5°x0.5° and 2°x2°."

**Line 173 are the 5,000 samples MODIS, MLS, or MODIS-MLS samples?**

> This number refers to MODIS-MLS samples. We changed the sentence accordingly: "While not every grid box contains the same number of profiles, each area contains at least 2,100 MLS-MODIS samples. A maximum in sample frequency is observed over the regions with denser MLS coverage around the poles."

> Note that we have changed the horizontal resolution from 60°x60° to 15°x15° in response to a comment from referee #2. We also added two separate maps, one for the statistics of the total MLS-MODIS data set, and one for the statistics of the clear and cloudy cases (as defined in section 3.2).

**Line 262. Approximately how many epochs are calculated?**

> When we started to test different setups, we ran each model with a fixed number of 10,000 epochs. However, we quickly noticed that each model starts to converge to a solution (i.e., the validation loss does not decrease any longer) much earlier. This number is comparatively low; indeed, more complex regression simulations performed by the

MLS group require 10-100 times more epochs. This means that the 2-class binning performed by the cloud classification models in this study is computationally inexpensive and only takes about 1 day.

We added this information to the manuscript: "Note that the lowest validation loss usually occurred after ~2,000-3,000 epochs for both the cloud classification and $p_{CT}$ prediction."

**Line 318 clarify what is meant by "classification going forward".**

We meant to say that in this study, we use the model with the highest Matthew's Correlation Coefficient ($Mcc$), out of the 100 we trained. One could think of other approaches, e.g., picking the one with the median $Mcc$, or another binary metric. However, since the validation scores are so close to each other, there really isn't a practical difference between each of the models.

We changed the sentence to:
"Given the statistical robustness of the results, the model with the highest $Mcc$ and lowest RMSD provide the ANN weights for cloud classification and $p_{CT}$ prediction in this study, respectively."

**Line 549. If the current MLS data version is V5, why not include the new ANN capability in the V5 product instead of "future versions of the v4.2x" product?**

The way we phrased the outlook was confusing. We compared the ANN cloud flag to the operational v4.2x cloud flag, as v5.x data was still being processed at the time of writing. The MLS radiances and cloud detection code are identical between the two versions, however, revisions to the atmospheric composition retrieval algorithms yield some subtle differences in the cloud status flags. These differences have no impact on the conclusions reported in this manuscript. Since the ANN cloud classification scheme only uses MLS radiances as input, it is independent of the MLS L2 algorithm version.

We plan to continue to provide both v4.2x and v5.x data products for the foreseeable future. In the revised manuscript we changed this sentence to:
"This new cloud classification scheme, which will be included in future versions of the MLS dataset, provides the means to reliably identify profiles with potential mid- to high-level cloud influence. Note that MLS radiances are not affected by the change from v4.2x to v5.0x."

We also added a clarifying statement to section 2:
"Note that the sampled radiances are identical between the two versions, while revisions to the atmospheric composition retrieval algorithms yield subtle differences in the derived cloudiness flags."

References:

Guillaume, A., Kahn, B. H., Yue, Q., Fetzer, E. J., Wong, S., Manipon, G. J., Hua, H., & Wilson, B. D. (2018). Horizontal and Vertical Scaling of Cloud Geometry Inferred from CloudSat Data, Journal of the Atmospheric Sciences, 75(7), 2187-2197. Retrieved Aug 27, 2021.

---

## Author Comment (AC2)

We'd like to thank the editor for handling our manuscript, as well as reviewer #2 for reading our manuscript and offering a large number of detailed suggestions and improvements.

We have carefully read through all the comments and questions and revised the manuscript accordingly. Please find our point-by-point response to reviewer #2 below. Here, the reviewer's general remarks, as well as the specific questions/comments, are formatted to be left-aligned text in bold font. Our responses are indented and formatted in regular font.

Here is a summary of the major changes in the revised manuscript:

1) We defined an independent test data set that is not included in the determination of hyperparameters or the training of the ANNs. The new splits are 70%/20%/10% for the training/validation/test data set. We also excluded the training data set from the evaluation of the model performance. This means that, e.g., the histograms in Figure 5 or the global maps in Figure 6 are generated by profiles the ANN has not been trained on. This allows for a fairer evaluation of ANN performance.

2) Since we removed profiles for the test data set and introduced new splits between training and validation data, we needed to repeat the k-fold cross validation and training of the ANNs. This turned out to be a necessary step, as we were able to fix three bugs in our algorithm setup: (i) We had not considered the number of hidden layers to be a hyperparameter. Tests revealed that models with only one hidden layer slightly outperformed those with two layers for the cloud classification scheme (the cloud top pressure models still use two layers). (ii) We had accidentally shuffled the training and validation data twice. While this had (obviously) no effect on training performance, it affected the correct recording of the respective profile indices. In other words, we did not correctly track the profiles in the training and validation data sets. This, in turn, means that the validation statistics presented in Figures 4 and 10 were inaccurate, as the presented "validation data" was actually comprised of random profiles from both the training and validation data set. Note that in the original manuscript version, model performance was evaluated for the combined training and validation data set (e.g., Figures 5 and 7), which means this mistake had no effect (i.e., the evaluation was based on a combined data set). (iii) The wrong control file provided the cloud top pressure model in the original manuscript version. That specific model had no weight decay (i.e., the model could learn training data very well, to the detriment of generalization) and early-stopping was turned off. This, together with the wrong recording of training and validation indices, resulted in the unrealistic correlation coefficients of 0.99. The model in the revised manuscript exhibits a much more reasonable correlation coefficient of 0.82.

3) We added more detailed explanations of machine learning terminology and descriptions of the considered hyperparameters.

4) We replaced one of the example scenes over South East Asia. In the original manuscript, the two scenes in Figure 9 looked very similar. Instead, we decided to present a more complex cloud field, which nicely illustrates the performance of the cloud classification model, while highlighting instances where the cloud top pressure prediction struggles.

5) We extended the analysis of the cloud top pressure ANN performance considerably. That section now includes additional statistical analysis of the difference between predictions and observations, as well as the model's ability to detect clouds <400, 350, and 300 hPa.

We also added global maps of the model performance, as well as comparisons between MODIS, ANN, and v4.2x data (similar to the cloud classification analysis). Example maps now contain the same scenes as for the cloud classification part.

**Overall, in my view there are three aspects of the paper that could have been addressed differently:**

**Dependance of the statistical inversion model on a given cloud properties dataset. I would say that this has not been properly emphasized in the paper. Retrieving cloud properties is not straightforward, and relatively large differences are found when comparing products. If the statistical model approximating MLS radiances and cloud properties would have been trained with, e.g., Calipso cloud properties, or even a previous version of the MODIS dataset, that would have resulted in a different model. This is not a criticism of the choice of the MODIS product, which is perfectly justified, but a reminder that this is an important part of the algorithm. For instance, biases on the cloud product are likely to be learned by the statistical model, and not be shown when evaluating the model performance with the same dataset used to train the model. Sentences like "This algorithm is designed to classify clear and cloudy conditions for individual MLS profiles, based purely on the sampled MLS radiances" do not help to convey this message.**

This point is well taken. The proposed models are only designed to reproduce the MODIS targets, which themselves exhibit their own retrieval uncertainties and biases. We certainly did not intend to portray our results as the actual, atmospheric truth.

In particular, we made sure to avoid phrases like "true values" or "truth" when we discuss the observations. To make this point more clearly, we have added sufficient disclaimers about the relationship between MODIS targets and the ANN estimates, while re-writing some sentences to convey the fact that the models only replicate the MODIS results.

This includes this paragraph in section 3.2:
"The reader is also reminded of the fact that the proposed ANN schemes will try to reproduce, as best as they can, the MODIS-retrieved cloud variables. Those parameters, however, have their own uncertainties and biases, and the ANN will inherently learn those MODIS-specific characteristics. As a result, the ANN predictions should not be considered the true atmospheric state. Instead, they represent a close approximation of the observed values in the colocated MLS-MODIS data set."

Also, in section 4.1 and 4.3 (amongst others):
"The analysis in section 3.4 indicates that the ANN setup can reliably reproduce the cloudiness conditions identified by the colocated MLS-MODIS data set."

"While the analysis in section 4.1 illustrates that the new ANN-based cloud classification can reliably identify cloudy profiles (based on the definitions in section 3.2), it is important to make sure that there is no latitudinal bias in the predictions…"

We also added the following statement to the "Summary and conclusions" section:

"It is important to note that the predicted cloud parameters do not represent the true atmospheric state. Since each ANN was trained on the colocated MODIS targets, it follows that they, at best, will replicate the respective MODIS results. The MODIS retrievals, however, are characterized by their own uncertainties and biases, which are subsequently learned and reproduced by the derived models. This means that analyses of ANN performance in this study only provide an evaluation of how well each model can replicate the colocated MODIS retrievals."

There are more examples, and we believe that these steps assure the reader of the fact that the ANN predictions are not the "truth".

**Description of the ANN. In my view, it seems a bit unbalanced in terms of the elements that are, and are not, described. On the one hand, most readers are likely to be familiar with the description of the architecture of a standard multi-layer perceptron, so Section 3.1 could have been considerably shortened. On the other hand, terms like L2 regularization, mini-batches, Nesterov momentum, less arguably familiar concepts, are used without any further explanations. I know that it is difficult to strike the right balance there, as depends on the reader, but I feel that this could have been done in a more appropriate way.**

We added numerous explanations to the revised manuscript. This comprises small additions like the following:

"In a feedforward ANN information only gets propagated forward through the different model layers and is not directed back to affect previous layers."

And:
"Generally, $F1$ assigns more relevance to false predictions and is more suitable for imbalanced classes, where the respective data sizes vary significantly."

We also re-wrote section 3.4, adding subsections and more descriptive text about the various hyperparameters:
"The hyperparameters to be determined are (i) the number of hidden layers, (ii) the number of neurons per hidden layer, (iii) the optimizer for the cloud classification, (iv) the mini-batch size, (v) the learning rate, and (vi) the value for the weight decay (i.e., the L2 regularization parameter). The number of hidden layers and neurons impact the complexity of the model. The choice of optimizer controls how fast and accurately the minimum of the loss function in Eq. (8) is determined, based on different feature sets and minimization techniques. During each iteration the model computes an error gradient and updates the model weights accordingly. Instead of determining the error gradient from the full training data set, our models only use a random subset of the training data (called a mini-batch) during each iteration. This not only speeds up the training process, but also introduces noise in the estimates of the error gradient, which improves generalization of the models. The learning rate controls how quickly the weights are updated along the error gradient. Thus, the size of the learning rate affects the speed of convergence (higher is better) and ability to detect local minima in the loss function (lower is better). Meanwhile, L2 regularization is one method to specify the regularization term $R$ in Eq. 8), where the sum of the squared weights is multiplied with the L2 parameter…"

However, the description of the model setup is very concise (about 1 page in the double-spaced draft format) and can easily be ignored if the reader is not interested in the details. For those interested in the specifics, and those who want to replicate the results, we think it important to include this short section. Therefore, we decided to leave this part in the revised manuscript.

**Retrieval of cloud top pressure. The paper gives the impression that the retrieval of cloud top pressure is not integrated in the paper from the beginning. While the cloud flag takes more of the paper, with different sections and comprehensive analyses, the cloud top pressure retrieval is not mentioned in the introduction (only mentioned at the end when describing the paper contents), the work takes just one section at the end of the paper, and seems much less detailed in terms of modeling and analyses. This may be intentional, as the authors may want to give more value to the cloud detection than to the cloud top pressure retrieval. But it feels a bit awkward, as if the cloud top pressure was an afterthought. The paper may have looked more consistent if, for instance, the ANN for the cloud top pressure would have also been described in the algorithms section, mentioning that these are 2 different inversion problems requiring different ANN setups (a binary classification, and a continuous mapping between 2 finite spaces), or if the qualitative assessments would have used the same selected scenes with a bit more of a joint discussion.**

As the reviewer discerned, the paper started as a pure cloud classification paper and later was extended, once the cloud top pressure ANN was developed. We have now changed the paper extensively to integrate the cloud top pressure estimates more naturally in the revised manuscript. We also extended the associated analysis and added more discussion.

This includes the abstract:
"Training a similar model on MODIS-retrieved cloud top pressure ($p_{CT}$) yields reliable predictions with correlation coefficients >0.82. It is shown that the model can correctly identify >85% of profiles with $p_{CT}$ <400 hPa. Similar to the cloud classification model, global maps and example cloud fields are provided, which reveal good agreement with MODIS results."

Also, the introduction:
"The performance of the subsequent cloud top pressure predictions is presented in section 5, which comprises an evaluation of the prediction performance and an assessment of the model's ability to detect high clouds (section 5.1), global maps (section 5.2), and four example scenes comparing the ANN predictions to the MODIS results (section 5.3).

We added information in the algorithm section:
"The model for the cloud top pressure prediction uses a simple pass-through of the neuron output to the output layer."
And:
"Conversely, the model for the cloud top pressure prediction minimizes the mean squared error."

Also, an extra paragraph on the most suitable hyperparameters in section 3.4:

"For the $p_{CT}$ prediction, two-layer models noticeably outperformed single-layer ones, as the drop in average $r$ was $> 0.01$. Again, the number of neurons had only a minimal impact on model performance, with variations in $r$ of $\approx 0.02$. However, models with 800–1000 neurons performed best, so we again set this number to 856. The best optimizer, learning rate, L2 parameter, and mini-batch size were found to be Adam, $10^{-4}$, $50e^{-4}$, and 1024, respectively."

We added the respective validation performance to Figure 4, illustrated in the new panels c and d.

We extended the associated analysis in section 5 considerably. This section now discusses statistical differences in more detail, as well as the model's performance at detecting high clouds for different cloud top pressure thresholds. The new section 5.2 presents similar maps of global predictions and *F1* scores, as well as comparisons to the MODIS observations and the current v4.2x flag.

We believe that these changes integrate the cloud top pressure retrieval more organically into the revised manuscript.

**Some more specific comments are given below. There are mainly about the statistical inversion, as the MLS team knows very well its instrument, and the discussion concerning the instrument capabilities, and what it can be retrieved from its radiances, is already very solid.**

**Specific comments**
**L58. "However, the reliance on estimated clear sky radiances and the use of predefined thresholds induces uncertainties in the current algorithm". This can be interpreted as a lack of confidence on the clear sky radiative transfer modelling, which is the basis of most of MLS retrieval work. I would say that the problem is more defining universal thresholds that can reliably identify the clouds, as illustrated later in the paper.**

Indeed, we did not mean to suggest any lack of confidence in the radiative transfer modelling. Instead, we wanted to convey the fact that fixed thresholds (which are defined rather conservatively) only allow identification of the thickest of the high-reaching clouds (see Figure 7d of our manuscript).

We changed the sentence as follows:
"However, the reliance on global, conservatively defined thresholds will inherently induce uncertainties in the current cloud detection scheme."

**L70. "This algorithm is designed to classify clear and cloudy conditions for individual MLS profiles, based purely on the sampled MLS radiances." I find this sentence misleading, as you always need something else than the radiances to do an inversion. The relationship between the radiances and your parameter of interest needs to be established, e.g, by a radiative transfer model (original cloud flag), or by a statistical model (proposed flag). For**

**this specific work, the statistical algorithm depends on the sampled MLS radiances and their relationship to the MODIS retrievals.**

> This sentence was meant to convey the fact that we only use MLS radiances AS INPUT for the algorithm, i.e., no ancillary data from a forward model or other instrument are required. In other words, these predictions don't require additional information (e.g., sensor zenith angle, geolocation, other MODIS parameters, simulated meteorological fields, etc.). Of course, we first need to establish the statistical model, which is needed to perform cloud classifications.
>
> We deleted the confusing part and the sentence now reads as follows:
> "This algorithm is derived from colocated MLS samples and MODIS cloud products and is designed to classify clear and cloudy conditions for individual MLS profiles."

**L72. "both high and mid-level clouds (e.g., stratocumulus and altostratus)". I know that the mentioned clouds are just examples, but they seem to coincide with mid-level clouds. It may give the impression that "cirro" clouds are not targeted, e.g., because of limitations of the MODIS cloud product.**

> This was purely an oversight, as the original examples went missing in the submitted manuscript. The MODIS retrieval for reasonably thick cirrus is quite reliable (obviously, we don't expect to detect subvisible cirrus with MLS radiances, nor do we expect those clouds to impact the MLS samples).
>
> We added the respective examples and the sentence now reads:
> "… detection of both high (e.g., cirrus and cumulonimbus) and mid-level (e.g., stratocumulus and altostratus) clouds, …"

**L75. "Aqua MODIS observations are ideal for this study". Perhaps "ideal" is not the best word here, given the large difference with MLS in terms of observing geometry, spatial resolutions, etc. Suitable?**

> Here, "Ideal" referred to the fact that (i) MODIS is installed on the Aqua spacecraft and thus provides the means for close temporal colocation, and (ii) the operational MODIS cloud data set provides reliable cloud products that cover the whole MLS mission period.
>
> Following the reviewer's suggestion we changed the sentence as follows: "Of the major satellite instruments, Aqua MODIS observations are most suitable for this study, as they provide operational cloud products on a global scale that are essentially coincident and concurrent with the MLS observations."

**L95. "The most recent MLS dataset is version 5; however, at the time the ANN was being developed, reprocessing of the entire 16-year MLS record with the v5 software had not yet been completed. Accordingly, L2GP cloudiness flags in this study are provided by the version 4.2x data products (Livesey et al., 2020), and v4.2x is also the source for the Level 1 radiance measurements used herein". Is there anything significantly different in the V5 radiances that could have an impact on this work? I guess not, but it may be worth commenting that.**

Indeed, this statement might be confusing for the reader. This also concerns the last paragraph in the Summary and Conclusions section. The MLS radiances and cloud detection code are identical between the two versions, however, revisions to the atmospheric composition retrieval algorithms yield some subtle differences in the cloud status flags. These differences have no impact on the conclusions reported in this manuscript. Since the ANN cloud classification scheme only uses MLS radiances as input, it is independent of the MLS L2 algorithm version.

We added the following sentence to section 2 in the revised manuscript: "Note that the sampled radiances are identical between the two versions, while revisions to the atmospheric composition retrieval algorithms yield subtle differences in the derived cloudiness flags."

We also added the following information to the "Summary and conclusions" section: "Note that MLS radiances are not affected by the change from v4.2x to v5.0x."

**L115. "Table 1 lists the 208 days that comprise the global data set used in this study. It consists of eleven random days from each year between 2005 and 2020, as well as a pair of two consecutive days to bring the yearly coverage to thirteen days." In my view Table 1 is not really needed, i.e., knowing that in 2012 d06 was day-of -year 169 is not critical information to pass to the reader.**

We respectfully disagree with the reviewer on this point. Stating the data set is an essential part of every observational study. Usually that is done by mentioning the years covered by a specific analysis (e.g., 2004-2013). However, due to the vast amount of data and computational time required for the MLS-MODIS colocation, we can only use individual days for our analysis. Therefore, to guarantee transparency and the reproducibility of our results, we feel that Table 1 is needed in the manuscript.

However, we concede that the majority of users are probably not interested in that information. Therefore, we moved the table to the Appendix. This means, the Table is clearly marked as ancillary information and most readers can ignore it, but those who are interested in the details can still look up the individual days.

**L118. "Particular attention was paid to ensure that each month is represented (close to) equally in the final data set". I am a bit puzzled here. Why not something as simple as randomly selecting one day per month and year? Then all months are equally represented without any need for further checks.**

In fact, this is almost exactly what we did. However, occasionally there are days in the MLS data record, where not all MLS observations yield useable data. This can be due to spacecraft maneuvers, calibration procedures, or instrument issues. While this doesn't happen often, we wanted to make sure not to include such days.

We concede that this might be confusing to the reader, and represents a detail we can safely omit from the manuscript. We therefore re-wrote this part to:

"It consists of twelve random days annually, one for each month, for the years between 2005 and 2020, as well as one additional day each year that forms a set of consecutive days. This brings the yearly coverage to thirteen days."

**L131. I was wondering about the choice of two hidden layers and not just one. Perhaps because this is a binary classification problem and the ANN search a decision boundary to separate clear and cloudy? But using the log-sigmoid function of the output node, the ANN will not be just separating into 2 classes, but outputting the probability of having a cloud. This could be interpreted as an ANN approximating a continuous mapping between two finite spaces, the radiances and the probability of having a cloud, i.e., a number between 0 and 1, and therefore one hidden layer may suffice.**

> The reviewer is absolutely right. One hidden layer is enough to map between the two classes with high accuracy. This simplifies the final model considerably (i.e., fewer variables to determine by the algorithm).
>
> As mentioned earlier, we did not consider the number of hidden layers as a hyperparameter in the original version of the algorithm and manuscript, and simply assumed that two layers would yield the best results.
>
> After restructuring the setup (70%/20%/10% splits between training/validation/testing data sets), fixing errors associated with the recording of validation and training data, and determining the most appropriate hyperparameters again (including the number of hidden layers), we indeed found that one-layer models yield the best performance scores. Here, average differences in $Ac$ are in the range of 0.005-0.01 (note that these differences, while consistently observed, are very small).
>
> We have changed the manuscript accordingly.

**L171. Why 60x60 degrees? A finer grid (e.g., 5x5 degrees) would be useful to better understand the geographical distribution of the samples. In fact, what is producing such an uneven distribution? Why the higher sampling over Africa?**

> Regarding the first part of this comment: The resolution was connected to Figures 7a and b, where we require sufficient sampling statistics to reliably determine the respective $F1$ scores (or other metrics, for that matter). We also wanted the spatial resolutions in the two Figures to be identical.
>
> We agree with the reviewer that an increase in spatial resolution captures the actual distribution better. After some testing, we found that a 15°x15° resolution (i) still yields sufficient statistics for the F1 calculation (>160 samples, from the validation and test data set, per grid box), and (ii) shows the distribution of data points in Figure 2 in more detail.
>
> Regarding the second part of this question: There are actually two data sets to consider here. The total number of profiles in the colocated MLS-MODIS data set are distributed quite evenly. The map in the original manuscript version, however, showed the distribution of clear sky and cloudy profiles, following the definitions in section 3.2. The uneven distribution is a direct result of these definitions. For example, the African

continent exhibits regions with very low and very high cloud cover (see Figure 7e-f). As a result, this specific (coarse) grid box contained more samples. Conversely, regions with ~50% cloud cover on average will contain fewer samples.

We added more panels to Figure 2, which now shows both the total number of profiles in the collocated data set and statistics for our cloudy and clear definitions. The two panels are shown below for convenience:

[Figure]

Fig. 1: Sample frequency for the (c) complete MLS-MODIS data set and (d) cloudy and clear profiles, following the definitions in section 3.2.

We also discuss the reasons why certain areas contain more/fewer samples.

**L185. "Naturally, these definitions leave some profiles undefined (e.g., those with C in the range 1/3–2/3)". The definition of clear and cloudy classes is perfectly justified, but excludes the undefined profiles from the training dataset, as properly stated in the text. Later on, when applying the ANN to classify the observations, the ANN will have to classify profiles similar to those unseen during the training. So, in principle, the ANN will be extrapolating. Could this be a problem? Could have been an alternative to train with all cases where pCT > 700 hPa, but targeting the continuous variable C with values in the range 0-1, instead of the two defined classes? The ANN would be similar, apart from choosing a loss function more appropriate for a continuous output space.**

This is an interesting suggestion. We actually tried to train a model, as suggested by the reviewer (albeit without going through the rigorous determination of hyperparameters). Unfortunately, this method doesn't seem to be able to replicate the MODIS observations.

[Figure]

Fig. 2: Predicted cloud cover (*C*) vs MODIS-observed *C*. Data are provided by the colocated MLS-MODIS data set.

While there is a general positive correlation for the relationship, the correlation coefficient is only $r$=0.14. It appears that MLS radiances can be successfully employed to detect clouds (and their associated pressure levels), but the sensitivity towards cloud fraction seems very limited.

As we demonstrate in the revised manuscript, the model performance for the validation and test data set is basically identical, and the performance for a modified classification is similarly high (as long as the observed cloud top pressure is below 700 hPa). If low clouds are also considered, model performance decreases significantly. However, that drop can be attributed to the number of false negatives, i.e., the model simply misses those low-level clouds. Since we don't expect MLS to be sensitive to those observations, we don't consider this to be a problem (it would be a problem, if the number of false positives, i.e., false cloudy predictions, increased).

Note that we also tried to predict total water path, similarly unsuccessfully (which is not surprising).

**L245. "Overall, the input matrix for the training and validation of the ANN is of shape 1,710 × 162,117". 1710 features are still a relatively large number. Even if a channel selection has been applied, one may wonder about the possibility of further reducing the dimensionality of the input space with some of the typical feature extraction techniques. Doing so is a common procedure to decrease computational burden and enhance the generalization properties of the ANN. I do not know well the MLS radiances, but I assume that there is some degree of correlation between the 1710 features, which could make possible this further dimensionality reduction.**

We tested this extensively in the beginning of the analysis. Specifically, we used Principal Component Analysis to reduce the dimensionality of the input matrix.

However, we found that, in order to explain 95% of the variability, we still needed to consider ~1,000 features.

Note that we are not concerned about the computational expense associated with the rather large dimensionality of the ANN. Training 100 models with our current setup takes ~1 day on our computing cluster, while predictions for individual days finish in less than 1 minute. Considering that we are not limited by computational constraints, we decided to keep the model as complex as possible (following the results of the *k*-fold cross validation) to ensure optimal classification performance.

We added information about the computational costs in the revised manuscript to inform the reader about the fact that the complexity of the ANN is not an issue:
"Training of these 100 models took ~1 day."

**L253. "The number of neurons per hidden layer is set to 856, which corresponds to the average between the number of nodes in the input and output layers". This sentence may make the reader think that this is a sort of standard procedure to fix the number of neurons, which is not. The number of nodes sets the capacity of the model to learn complex mappings, with small (large) ANNs having the risk to underfit (overfit). But the number of nodes is not typically considered as an hyperparameter to control model complexity. Instead, training techniques similar to the one applied in the paper are applied to control the generalization capacity of the ANN.**

Our explanation was not sufficient. We did determine the number of neurons in a manner similar to that of the other hyperparameters. We varied this parameter between 100 and 1200, in increments of 100 neurons per layer. Starting from about 400 neurons per hidden layer, the observed changes in *Ac* are in the range of 0.0002. Nonetheless the maximum *Ac* value is observed for 800 neurons per layer, closely followed by models with 900 neurons per layer.

We did not think it useful to test performance for increments <100 neurons per layer. Again, the differences in derived *Ac* are very small. As the maximum was supposedly somewhere between 800 and 900 neurons, we simply chose to use the average between the number of features (*nf*) and labels (*nl*) to determine the final setup. As the reviewer mentions, this is not a standard procedure, although a considerable number of discussion posts found online do recommend to set this value to 2/3*(*nf*+*nl*) or 1/2*(*nf*+*nl*). For us, it is just a convenient coincidence that the average between *nf* and *nl* is in the region of maximum *Ac*.

This is the corresponding statement in the revised manuscript:
"The number of neurons per hidden layer had a negligible impact, as long as the number was larger than 200. However, the models with 800 and 900 neurons exhibited average *Ac* values that were 0.0002 higher than those of other setups. We ultimately set it to 856, which corresponds to the average between the number of nodes in the input and output layers (i.e., 1,710 and 1, respectively)."

**L253. The 856 nodes in the hidden layer results in a very large ANN. If you are using a fully connected ANN of the type described, the number of weights to be adjusted during the training is 1710 (inputs) x 856 (hidden layer one) + 856 x 856 (hidden layer 2) + 856 x 1(output), and then you have 856 + 856 + 1 biases. This is around 2 million of model parameters, even larger than the number of samples x input features in the training dataset. I may be missing something, but it is hard to believe that a simpler ANN cannot be setup to classify the radiances into the two clear and cloudy cases.**

The reviewer is correct; a simpler model absolutely suffices, and classification performance, while reduced with a simpler setup, is still high. However, the highest $Ac$ values are still observed for ~800 neurons per hidden layer. In our testing, the spread in derived $Ac$ was in the range of 0.015 (as long as the number of nodes was >200 and L2 parameter <$10^{-3}$. The lowest value of $Ac$~0.78 was recorded for 100 neurons per layer, with an L2 parameter of $10^{-1}$.

Reducing the number of hidden layers from 2 to 1 reduces the number of model parameters considerably, from 2,199,922 to 1,466,330.

We changed the manuscript accordingly.

**L255. "(ii) the learning rate, (iii) the mini- batch size, and (iv) the value for the weight decay (i.e., the L2 regularization parameter)". What do you mean by mini-batch? What is L2 regularization? Square of the weights instead of absolute values?**

This is one of the big changes in the revised manuscript, as we added a lot of extra information about the different hyperparameters in section 3.4:
"The hyperparameters to be determined are (i) the number of hidden layers, (ii) the number of neurons per hidden layer, (iii) the optimizer for the cloud classification, (iv) the mini-batch size, (v) the learning rate, and (vi) the value for the weight decay (i.e., the L2 regularization parameter). The number of hidden layers and neurons impact the complexity of the model. The choice of optimizer controls how fast and accurately the minimum of the loss function in Eq. (8) is determined, based on different feature sets and minimization techniques. During each iteration the model computes an error gradient and updates the model weights accordingly. Instead of determining the error gradient from the full training data set, our models only use a random subset of the training data (called a mini-batch) during each iteration. This not only speeds up the training process, but also introduces noise in the estimates of the error gradient, which improves generalization of the models. The learning rate controls how quickly the weights are updated along the error gradient. Thus, the size of the learning rate affects the speed of convergence (higher is better) and ability to detect local minima in the loss function (lower is better). Meanwhile, L2 regularization is one method to specify the regularization term $R$ in Eq. (8), where the sum of the squared weights is multiplied with the L2 parameter:

$$R = L2 \cdot \sum \omega^2 + \varpi^2 + \Omega^2 \text{ (9)}$$

Note that for clarity we omitted the indices for the weights in Eq (8). The amount of regularization is directly proportional to the value of the L2 weight decay parameter. Regularization usually improves generalization of the models. More information about

ANN hyperparameters and their impact on the reliability of model predictions can be found in, e.g., Reed and Marks (1999) and Goodfellow et al. (2016)."

**L256. If a weight decay term is used, should not have been included in the loss function of Eq. 8? That will make clear what L2 means.**

We added a regularization term ($R$) to Eq. (8). In the "Training and validation" section (section 3.4) we then describe that regularization term in more detail (see previous answer).

**L257. "of the cost function in Eq. (8)". Perhaps saying loss function, to refer to Eq. 8 with a single name throughout the paper?**

We changed this to "loss function".

**L262. "Instead, the models are run with a large number of epochs, and the lowest validation loss is recorded, so an increase in validation loss during the training (i.e., cases where the model is overfitting the training data at some point) has no impact on the overall performance evaluation". It is not clear to me how you apply an "early-stopping technique" to control model complexity here. There can be situations where the validation loss starts to be smaller than the training loss, i.e., an indication of over-fitting, but, nevertheless, it keeps decreasing, although with a smaller rate than the training loss. At some point, the validation loss may reach a minimum. Is that minimum the "lowest validation loss" you describe, where you consider your ANN properly trained, so those are the selected ANN weights? But it could be the case that at that point the training loss was already smaller that the validation loss for a large number of epochs, so in principle the ANN could be already overfitting. Perhaps it is just not well explained, or I am missing something.**

There is a very small gap between the training and validation loss at the point of the minimum. However, for reasonably complex models this is to be expected.

As mentioned in the revised manuscript, we tested different model setups with varying degrees of regularization (we also tested models with Gaussian noise applied to the layer outputs). For large amounts of regularization, the small gap between training and validation loss disappears. However, these models also consistently exhibited the lowest performance scores. For example, changing the weight decay term from $10^{-4}$ to $10^{-2}$ mitigated the gap between training and validation loss. However, it also decreased $Ac$ by 0.05 (for the cloud classification models) and the correlation coefficients by 0.04 (for the cloud top pressure models). Similarly, qualitative assessment of model performance by means of example maps (like those presented in Figures 8 and 9 in the manuscript) confirmed the poorer performance.

Regarding the impact of the "early-stopping", we found that the validation loss never really increased, even for a large number of epochs. This is true for both model setups (using the final set of hyperparameters). The validation loss basically converges and stays at a global minimum. We confirmed this by running models without "early-stopping", and those performed almost identically to the reported models.

We added this information to the revised manuscript: "Note that the lowest validation loss usually occurred after ~2,000–3,000 epochs for both the cloud classification and $p_{CT}$ prediction. No obvious increase in validation loss was observed, even for a large number of epochs.

**L273. "The ideal setup for the ANN". Perhaps "ideal" is not the best word here, as there will exist a number of ANNs performing very closely (e.g., close but different number of nodes). An "appropriate" setup for the ANN?**
We changed it according to the reviewer's suggestions.

**L288. "This analysis revealed that the stochastic gradient descent optimizer, using a learning rate of 0.001, and a Nesterov momentum value of 0.9 yielded the overall best validation scores. The best weight decay and mini-batch size values were found to be 5 × 10−4 and 1024 (i.e., 0.8% of the training data), respectively." Stochastic gradient descent? Nesterov momentum?**
This part is not included in the revised manuscript. As mentioned earlier, we revised the ANN training setup (split of the data set into 70%/20%/10% training/validation/test data, correct assignment of the profile indices, treating the number of hidden layers as a hyperparameter). After re-running the $k$-fold cross validation with this setup, we found that the Adam optimizer yields (slightly) higher performance scores.

The revised manuscript now states the following:
"The Adam optimizer with a learning rate of $10^{-5}$ yielded the overall best validation scores for the cloud classification. Note that we applied the Adam optimizer with the standard settings described in the "Keras" documentation."

**L292. "By chance, the most obvious cloud cases (e.g., C = 1 and very large QT values) might have ended up in the validation data set, or vice versa, and the trained weights might be inappropriate". I would say that your random selection of training and validation cases, together with the relatively large number of samples, makes this very unlikely to happen. If it is happening, I would revise the sample selection strategy.**
We do not fully agree with the reviewer on this point, although we concede that the described scenario is rather unlikely (and, obviously, did not happen in our study). However, had we only developed a single model, even with a large data set, our confidence in the reliability of the derived weights would have been considerably lower. Only after training a large number of models and confirming that the performance scores were virtually identical, were we assured that the derived models were reliable.

As a result, we decided to keep that statement in the revised manuscript.

**L294. "Moreover, a large disparity in validation scores for multiple models might be indicative of an ill-posed problem, where the MLS observations do not provide a reasonable answer to the cloud classification problem". Yes, this is for me the valid reason to undertake these tests.**
Thank you.

**L297. "In this study, 100 different models are developed ". To be clear, the only thing that changes is the split in the training-validation datasets, the model hyperparameters are set to your final configuration, right?**

Correct. The same model setup, but a different random split between training and validation data.

We added the following sentence to the revised manuscript: "The hyperparameters are identical for each model."

**L300. "The output of each ANN model is a cloudiness probability (P ) between 0 (clear) and 1 (cloudy)". Perhaps this should have been already mentioned earlier in the text, e.g., around Eq. 7, as this is the consequence of building the binary classifier with a softmax function in the output node.**

We changed the respective sentence in section 3.1 (right before stating our choice of activation function): "We aim for a binary, two-class classification setup (i.e., either cloudy or clear designations) and information about the probability for each predicted class. As a result, the softmax function..."

**L339. Perhaps there is no need to give both absolute number of cases and percentages of the total. I personally found the percentages more informative.**

We removed the absolute numbers and only mention the percentages in the revised manuscript.

**L345. "Only 1.7% of clear profiles are falsely classified as cloudy by the new ANN algorithm, while the current v4.2x status flag mislabels 6.2% of these profiles". I do not have doubts that the new cloud flag performs better than the V4.2 one. But I think it is worth mentioning that the V4.2 flag will always be penalized in these comparisons. The new flag has been trained on the same dataset used for the evaluation, while the V4.2 is independent of that dataset. For instance, biases in the MODIS cloud properties are likely to be learned by the ANN, and not shown in the evaluation. The V4.2 cloud, on the contrary, knows nothing about those biases.**

Indeed, the ANN will learn the idiosyncrasies of the MODIS data set, and the current v4.2x flag will always be penalized. Additionally, the L2 flag was not designed to detect all kinds of clouds, but only the thickest ones that reach high enough into the upper troposphere.

Due to the revised analysis the number of false positives (i.e., cloudy predictions for clear profiles) is now closer to the v4.2x results. Therefore, the difference in classification performance between the ANN and v4.2x can be almost exclusively explained by their ability to detect clouds. We mention in the manuscript that the current flag is designed to only detect high-reaching convection, for which associated clouds are sufficiently opaque (note that the ANN appears to perform better for both the very opaque and the high clouds). It is unlikely that the poor v4.2x performance can be explained by uncertainties and biases in the MODIS retrievals. After all, the MODIS data set, while not indicative of the atmospheric truth, is widely considered to be one of the most reliable sources of cloud

information. We can safely assume that the MODIS results are a good approximation of the actual cloud conditions.

However, we agree that it is still important to add a disclaimer to the revised manuscript (in addition to the other discussion concerning the fact that the MODIS results are not the actual atmospheric truth). We added the following statements at the beginning of sections 4 and 5:
"Note that a comparison between v4.2x and ANN results will naturally favor the ANN predictions, particularly any comparison made with reference to MODIS observations. Evaluating the performance of each cloud flag is based on the respective agreement to the MODIS-observed cloud conditions. However, the ANN is designed to replicate the MODIS results, while the v4.2x algorithm is not aware of the MODIS data set (including its uncertainties and biases)."
And:
"Similar to the cloud classification analysis, a comparison between v4.2x and ANN prediction performance will favor the ANN results, since the ANN is designed to replicate the MODIS observations."

**L358. "It is essential to understand the ANN performance for the undefined, in-between cases". Yes, I fully agree. As mentioned above, the ANN has never seen the undefined cases, so in principle it is extrapolating to classify those profiles.**

While the reviewer is correct, the ANN also has never learned the validation and test data (although profiles in the validation data set were used to determine the best set of hyperparameters). As stated in our reply to the prior comment (and demonstrated in the manuscript), the ANN provides good performance for previously unseen data.

If the observed cloud top pressure is below 700 hPa, the performance scores are only slightly lower than for the validation and test data set. For example, the *F1* score for the in-between cases is 0.90, while it is 0.94/0.96 for the validation/test data. This indicates that, even though the model is extrapolating to classify those samples, it does a reasonably good job. No changes to the text were made in response to this comment.

**L379. "Due to the looser definitions, there is a significant drop in performance scores". Could it be not just the looser definitions, but the fact that now the classes include cases never seen by the ANN as they were not part of the training dataset? Perhaps this is a more realistic assessment of how the ANN will perform later when faced with all MSL radiances.**

As mentioned earlier, this is definitely not the case. The revised manuscript now summarizes the classification performance for the validation and test data sets. All profiles from the training data are excluded from the analysis. The results for the undefined cases, which are also presented in the revised manuscript, are very close to the performance scores of the training/validation data.

**L399. "Each grid box covers an area of 60°x60° (latitude and longitude)". Why not showing in a finer grid? For instance, the 3°x5° used for the remaining plots.**

In order to reliably calculate the binary statistics (e.g., *F1* score or *Mcc*), we not only need to collect enough data points per grid box, but also the denominator in Eqs. (11-13)

needs to be non-zero. Figure 3 of this reply shows the results for the *F1* score, if we use grid boxes with a 3x5 degree spatial resolution:

[Figure]

Fig.3: *F1* scores derived from the ANN (left panel) and v4.2x algorithm (right panel). Each grid box covers an area of 3°x5°.

Note the large number of grid boxes for which the calculation of the *F1* score fails, especially for the v4.2x algorithm. Here, there are either not enough data points or the denominator in Eq. (12) becomes 0. We only use the validation and test data set for this analysis. This is not a problem for the other panels in Figure 7 (panels c-f), as we use multiple years to calculate those statistics.

However, similar to the maps in Figure 2, we changed the resolution to 15x15 degrees. This ensures that we have >160 profiles per grid box and, apart from a few grid boxes in the v4.2x analysis (mostly in the high latitudes), we can always calculate the *F1* score. The respective maps are included in this reply for convenience:

[Figure]

Fig. 4: Like Fig.3, but now each grid box covers an area of 15°x15°.

**L399. "In contrast to the results for the ANN algorithm, there is a clear latitudinal dependence for the performance of the v4.2x algorithm". As I mentioned above, these comparisons penalize v4.2 as we are evaluating not with true cloud properties, but with the MODIS cloud properties targeted by the new flag. For instance, there may be the case that there is a latitudinal bias in the MODIS cloud parameters, and the ANN may learn it. But as he F1 scores are so poor for v4.2, we can still conclude that the new flag outperforms the v4.2 flag on the basis that MODIS C6 provides a realistic representation of clouds, even if not free from errors.**

It is very unlikely that the difference in classification performance can be explained by the fact that the ANN learned any MODIS uncertainties or biases. Additionally, the shortcomings in the v4.2x predictions can be well explained.

However, as mentioned in the reply to an earlier comment, we added disclaimers at the beginning of section 4 and section 5 that remind the reader of the fact that the ANN flag will be naturally favored in these comparisons. Also, the analysis in the revised manuscript is now only based on the validation and test data, i.e., the training data is excluded from the statistics.

Note that, since the MODIS cloud retrievals have an excellent reputation, we consider it a success of this study that we can (reasonably) replicate the MODIS observations.

**L429. "Due to the reduced sensitivity towards such clouds (see the discussion in section 3.3), the cloud covers predicted by the ANN are much closer to the MODIS results 430 that do not include low clouds". This was a nice test, with a very reasonable agreement.**
Thank you.

**L442. "Figure 8 shows two example cloud fields over the North American monsoon region. Nice and very illustrative examples.**
Thank you.

**L480. "The input layer and the two hidden layers remain unchanged from the cloud classification setup. The labels in the output layer, instead of being set to either "0" or "1" (i.e., clear sky or cloudy), now contain the respective pCT reported by the colocated MLS-MODIS data set". Two hidden layers are probably not required here, as this is definitely a continuous mapping between two finite spaces, so one hidden layer has the capability to approximate this mapping. Because of that, the ANN may look even more over-dimensioned than when acting as a binary classifier.**
While we agree that this is surprising, the two-layer models consistently outperformed one-layer models in our tests. If all other parameters remain fixed, the difference in correlation coefficient between one- and two-layer models is consistently in the range of 0.01.

This change is certainly small (about half the size of the drop in correlation coefficient associated with decreasing the number of neurons per layer). In fact, for the cloud top pressure models the lowest correlations were found to be ~0.75 (recorded for 100 neurons and a L2 parameter of $10^{-1}$) and even those models performed reasonably well.

However, as stated earlier, we are not concerned about the complexity of the model and the associated computational costs. With our current setup, training 100 models takes ~1 day on our computing cluster, while the predictions for a full day take <1 min. It is simply not necessary to reduce the complexity of the models, even though much simpler setups exhibit similarly high performance scores. No changes to the text were made in response to this comment.

**L483. "Similarly, the model optimizer, learning rate and mini-batch size reported in section 3.4 for the cloud classification ANN provide the best set of hyperparameters". Does it mean that you are using exactly the same hyperparameters? That looks strange to me, as you have a different mapping to approximate, so the optimal hyperparameters may not be necessarily the same.**

> As stated earlier, this was a mistake. We not only fixed an unfortunate mix-up between training and validation data, we also defined an independent test data set (which required a new split of 70%/20%/10% for the training/validation/test data). We also added the number of hidden layers as a hyperparameter and correctly applied early-stopping.
>
> As a result, the number of hidden layers and learning rate differ from the cloud classification model, as described in the revised manuscript.

**L484. "here the only change concerns the weight decay parameter, which is turned off." Is there a reason for that? Why approximating the new mapping does not require a weight-based regularization of the loss function, while the previous one required one?**

> Again, this was a mistake. The model read the wrong control file (which summarizes the model parameters). This has been corrected in the revised manuscript.

**L487. "Joint histograms of true (in the sense that they are the prescribed labels to train the ANN)". This is a good reminder about the fact that true here means MODIS-retrieved. It would have been nice to also introduce a bit more this idea when evaluating the results of the previous ANN with MODIS-retrieved cloudiness.**

> We made sure that the revised manuscript does not contain phrases like "true values" or "truth" when referring to the MODIS targets. We also added paragraphs in section 3.2 and the "Summary and conclusions section" that discuss the fact that the ANN will learn the MODIS uncertainties and biases.
>
> That particular sentence now reads as follows:
> "Joint histograms of observed and predicted $p_{CT}$ for all cloudy profiles in the validation and test data set are presented in Figures 10a and b, respectively."

**L488. "are presented in Figure 10". This seems like a much less detailed analysis of the ANN performance when retrieving pCT, compared with the cloud flag analysis. Even if the goal is only to differentiate between mid-to-low level clouds and high-reaching convection, other metrics than just the correlation, and how those metrics may depend on cloud type, altitude, location, and so on, could have been assessed.**

> We extended the cloud top pressure analysis considerably, by adding more discussion about the statistical differences between observed and predicted results and the ability to detect high clouds (section 5.1). We also added global maps that illustrate the model performance and present average distributions of the predicted values, similar to those presented for the cloud classification algorithm (section 5.2).

**L494. "Three example scenes with the MODIS pCT". Not suggesting that the presented cases are not interesting, but another option could have been to reuse the cases presented for the cloud flag. Specially the second example there seem to have the right mixture of**

**high and low clouds to permit an analysis of the pCT retrieval. Likewise, it could have been interesting to see the performance of the cloud classifier on the pCT scenes.**

We agree that it is best to present examples of cloud classification and cloud top pressure for the same example scenes. We changed Figure 12 accordingly.

**L515. "In this study, we present an improved cloud detection scheme based on the popular "Keras" Python library for setting up, testing, and validating feedforward artificial neural networks (ANNs)". Perhaps more interesting than mentioning the library would have been to briefly describe the setup, i.e., something like a "standard multilayer perceptron configured to act as a binary classifier by using a softmax activation function in the output node and a cross-entropy loss function to derive the weights".**

The revised manuscript now states the following:

"In this study, we present an improved cloud detection scheme based on a standard multilayer perceptron, a subcategory of feedforward artificial neural networks (ANNs). It applies a softmax activation function in the output layer for binary classifications (i.e., clear sky or cloudy), while a log–loss function is minimized to determine the model weights. A second setup, which applies a linear output in the output layer and determines the model weights by minimizing the mean squared error, is used to produce a cloud top pressure ($p_{CT}$) estimate from MLS radiances that approximates the MODIS retrievals."

**L533. "A comparison with the current v4.2x status flags reveals that for the complete data set in this study the new ANN results provide a significant improvement in cloud classification". Perhaps a good place to briefly mention than the "truth" here is the target of the ANN calibration used to derive the new cloud flag.**

We added the following paragraph before discussing the results:

"It is important to note that the predicted cloud parameters do not represent the true atmospheric state. Since each ANN was trained on the colocated MODIS targets, it follows that they, at best, will replicate the respective MODIS results. The MODIS retrievals, however, are characterized by their own uncertainties and biases, which are subsequently learned and reproduced by the derived models. This means that analyses of ANN performance in this study only provide an evaluation of how well each model can replicate the colocated MODIS retrievals."

**L550. "in future versions of the MLS v4.2x". This may seem confusing, as a V5 has already been mentioned in the text. Or perhaps v4.2x does not supersede v5 and both will be coexisting, with both v4.2x and v5 benefiting from the new cloud flag algorithm?**

The way we phrased the outlook was confusing. We compared the ANN cloud flag to the operational v4.2x cloud flag, as v5.x data was still being processed at the time of writing. However, while the change in version includes revisions to the atmospheric composition retrieval algorithms, the MLS radiances and cloud detection code are identical between the two versions. This means that the ANN cloud classification is independent of the MLS L2 algorithm version.

We plan to continue to provide both v4.2x and v5.x data products for the foreseeable future. In the revised manuscript we changed this sentence to: "This new cloud classification scheme, which will be included in future versions of the MLS dataset,

provides the means to reliably identify profiles with potential mid- to high-level cloud influence. Note that MLS radiances are not affected by the change from v4.2x to v5.0x."

---

## Referee Report (RR1)

Review of the revised paper "Improved cloud detection for the Aura Microwave Limb Sounder: Training an artificial neural network on colocated MLS and Aqua-MODIS data" by Frank Werner et al.

General comments

This revised paper nicely illustrates that the implementation of machine learning to MLS cloud classification leads to an impressive improvement in MLS cloud detection, compared to current operational techniques.

The authors have responded admirably to the comments written in the first review. The revised paper reads very well, and only minor suggestions are contained in this second review.

The paper should be published following very minor suggested changes and clarifications.

Very minor suggestions

Line 34. For some reason my copy of the paper has "Di Girolamo" outside of the right block margin.

Line 81 change to "four example scenes"

After equation (3) identify $\omega$ as a weight. Though the symbol is identified in Figure 1, as a reader I personally like to have symbols identified (and defined) also in the main text the first time they are encountered in the text.

Line 145 replace "These values" with specifically named symbols.

Line 265-269 I stumbled over the band counting in the current text. Should "three different bands" be replaced by "nine different bands"? The use of "For most of the ten bands" (instead of "For most bands" on lines 268-269) would clearly tell the reader that a total of ten bands are utilized.

---

## Author Response (AR2)

We again would like to thank the editor for handling our manuscript, as well as reviewer #1 for reading our manuscript a second time.

Please find our point-by-point response to reviewer #1 below. Here, the reviewer's general remarks, as well as the specific questions/comments, are formatted to be left-aligned text in bold font. Our responses are indented and formatted in regular font.

**General comments:**

**This revised paper nicely illustrates that the implementation of machine learning to MLS cloud classification leads to an impressive improvement in MLS cloud detection, compared to current operational techniques.**
**The authors have responded admirably to the comments written in the first review. The revised paper reads very well, and only minor suggestions are contained in this second review.**
**The paper should be published following very minor suggested changes and clarifications.**

**Very minor suggestions:**

**Line 34. For some reason my copy of the paper has "Di Girolamo" outside of the right block margin.**
> We fixed this issue.

**Line 81 change to "four example scenes"**
> Fixed, following the reviewer's suggestion.

**After equation (3) identify ω as a weight. Though the symbol is identified in Figure 1, as a reader I personally like to have symbols identified (and defined) also in the main text the first time they are encountered in the text.**
> We added the following description to the revised manuscript: "Here, the weights $\omega$ connect the observed brightness temperatures (and the bias) to the neurons in the first hidden layer."
>
> Similarly, we added these descriptions:
> "…, where the weights $\varpi$ connect the neuron output from the first hidden layer, as well as the bias, to the neurons in the second hidden layer. "
>
> And: "… connected to the single vector $\mathbf{L}$ in the output layer via weights $\Omega$."

**Line 145 replace "These values" with specifically named symbols.**
> Changed, following the reviewer's suggestion.

**Line 265-269 I stumbled over the band counting in the current text. Should "three different bands" be replaced by "nine different bands"? The use of "For most of the ten bands"**

**(instead of "For most bands" on lines 268-269) would clearly tell the reader that a total of ten bands are utilized.**

We changed these sentences to:

"Instead, ten different bands are chosen in total. Those are bands 2, 3, 6; bands 7, 8, 33; and bands 10, 14, 28 for the 190, 240, and 640 GHz spectral regions, respectively."

And: "For most of the ten bands, every second channel… "

We again would like to thank the editor for handling our manuscript, as well as reviewer #2 for reading the manuscript a second time.

Please find our response to reviewer #2 below. Here, the reviewer's general remarks, as well as the specific questions/comments, are formatted to be left-aligned text in bold font. Our responses are indented and formatted in regular font.

**The authors have done a good job addressing my major concerns, and the paper is more balanced in terms of presenting results for both variables investigated. Still, I'm left with the feeling that the paper could have been made written in a more concise form to facilitate its reading, but the authors have chosen to report their technical setup and findings with a great level of detail. In my view it is ready for publication, apart from some very minor technical corrections (e.g., L623 "As mentioned in the 1", or L625 HNO3 (e.g., ??)" that can be corrected while proof-reading the manuscript.**

> We compiled the Latex file again, which fixed the missing reference. "As mentioned in the 1…" was changed to "As mentioned in the introduction…"

> We carefully reviewed the manuscript again and checked for similarly misplaced labels and missing references.